# Multi-step processing of replication stress-derived nascent strand DNA gaps by MRE11 and EXO1 nucleases

Anastasia Hale[1,2], Ashna Dhoonmoon[1,2], Joshua Straka[1], Claudia M. Nicolae [1] ✉ & George-Lucian Moldovan [1] ✉

Accumulation of single stranded DNA (ssDNA) gaps in the nascent strand during DNA replication has been associated with cytotoxicity and hypersensitivity to genotoxic stress, particularly upon inactivation of the BRCA tumor suppressor pathway. However, how ssDNA gaps contribute to genotoxicity is not well understood. Here, we describe a multi-step nucleolytic processing of replication stress-induced ssDNA gaps which converts them into cytotoxic double stranded DNA breaks (DSBs). We show that ssDNA gaps are extended bidirectionally by MRE11 in the 3′−5′ direction and by EXO1 in the 5′−3′ direction, in a process which is suppressed by the BRCA pathway. Subsequently, the parental strand at the ssDNA gap is cleaved by the MRE11 endonuclease generating a double strand break. We also show that exposure to bisphenol A (BPA) and diethylhexyl phthalate (DEHP), which are widespread environmental contaminants due to their use in plastics manufacturing, causes nascent strand ssDNA gaps during replication. These gaps are processed through the same mechanism described above to generate DSBs. Our work sheds light on both the relevance of ssDNA gaps as major determinants of genomic instability, as well as the mechanism through which they are processed to generate genomic instability and cytotoxicity.

Maintenance of genomic stability requires the coordinated activities of a large number of proteins to detect and repair DNA lesions. When this coordination is lost, unrestricted engagement of repair factors may have detrimental outcomes on DNA integrity. This is best exemplified by the complex role of the BRCA pathway in genome stability. By loading RAD51 molecules on DNA, BRCA1 and BRCA2 control the extent of DNA resection by nucleases such as MRE11, DNA2 and EXO1[1–6]. In BRCA-mutant cells, these nucleases become hyperactive and degrade the DNA extensively resulting in genomic instability, and potentially also in hypersensitivity to genotoxic agents, including chemotherapeutic agents such as platinum-derived compounds and PARP1 inhibitors (PARPi)[5–10].

The DNA protection activity of the BRCA pathway is critical during DNA replication, when the replication machinery encounters unrepaired DNA lesions which block the progression of DNA polymerases leading to replication fork arrest. Arrested forks can be reversed by DNA translocases that anneal the nascent strands of the two sister chromatids, stabilizing the fork and allowing restart of DNA synthesis on the reversed arm[8,9,11]. However, unless protected through loading of RAD51 by the BRCA pathway, the reversed arm is nucleolytically degraded by MRE11, EXO1, and other nucleases, resulting in genomic instability and chemosensitivity[1–10].

Arrested forks can also be restarted by initiating de novo DNA synthesis downstream of the lesion, upon repriming by factors such as the primase-polymerase PRIMPOL and, on the lagging strand, the Polα-primase complex[10,12–15]. This leaves behind a short single stranded DNA (ssDNA) gap, to be filled at a later time, through DNA synthesis by polymerases including POLQ[16–18] and translesion synthesis (TLS)

---

[1]Department of Biochemistry and Molecular Biology, The Pennsylvania State University College of Medicine, Hershey, PA 17033, USA. [2]These authors contributed equally: Anastasia Hale, Ashna Dhoonmoon. ✉e-mail: cmn14@psu.edu; glm29@psu.edu

polymerases such as REV1[15,19]. The BRCA pathway may also participate in ssDNA gap filling using the nascent strand of the sister chromatid as template, through catalyzing a homologous recombination (HR) reaction behind the replication fork[20,21]. Reflecting the involvement of these multiple pathways in ssDNA gap metabolism, ssDNA gaps accumulate in cells with BRCA mutations, PRIMPOL overexpression, TLS inactivation, or POLQ knockdown[13,15–19,22–26].

While the inhibitory activity of the BRCA pathway against MRE11-mediated degradation of reversed forks has been extensively documented, recent studies showed that inhibition of the MRE11 exonuclease activity also suppresses nascent strand ssDNA gap accumulation in BRCA-deficient cells[15,27], suggesting that MRE11 exonuclease extends ssDNA gaps if they are not timely filled. MRE11 is a multi-domain nuclease which has both 3′−5′ exonuclease and endonuclease activities. It is thus conceivable that it would only be able to expand the gap in one direction, from the 3′ end. Whether the 5′ end of the gap is also subjected to exonuclease-mediated resection is unclear.

We recently identified the 5′−3′ exonuclease EXO1 as a critical component of the machinery that degrades reversed replication forks in BRCA-deficient cells[27]. We showed that fork degradation is initiated by the endonuclease activity of MRE11, which creates a nick in the double stranded DNA (dsDNA) region of the reversed arm. This nick is engaged bidirectionally by MRE11 which employs its 3′−5′ exonuclease activity to degrade the strand towards the DSB end of the reversed arm and by the 5′−3′ exonuclease activity of EXO1 for long-range resection towards the fork junction and beyond. Whether EXO1 also plays a role in ssDNA gap processing in BRCA-deficient cells is unclear.

Nucleolytic degradation of nascent DNA at stressed replication forks is a molecular process of great relevance to clinical oncology since a number of recent studies have proposed that it may regulate the response of cancer cells to genotoxic chemotherapy. Protection of reversed replication forks against nucleolytic degradation was proposed as a main contributor to chemoresistance of BRCA-mutant tumors[5–10]. More recently, ssDNA gap accumulation was found to better correlate with chemosensitivity in certain genetic backgrounds, suggesting gap suppression as the main mechanism of chemoresistance in these tumors[15,19,22–26]. On the other hand, BRCA2 separation-of-functions mutants which are proficient for HR but defective in fork protection and gap suppression show reduced chemosensitivity compared to BRCA2 mutants defective in all three activities, suggesting that BRCA2 promotes therapy resistance primarily through HR[28,29]. While by themselves ssDNA gaps are not considered cytotoxic, their accumulation has been associated with formation of cytotoxic double strand DNA breaks (DSBs), although it is a matter of debate if these DSBs are formed directly from the ssDNA gaps or arise subsequently during apoptosis[21,24,26,30]. Unless repaired by HR, DSBs can also cause genomic rearrangements, representing a main driver a chromosomal instability[31].

On the other hand, the extent to which DNA degradation contributes to genomic instability during carcinogenesis is less clear. The BRCA pathway is a major tumor suppressor pathway, with genomic instability thought to underlie carcinogenesis in BRCA mutation carriers[32]. While the relevant genotoxic exposure causing genomic instability in breast and ovarian tissues of BRCA mutation carriers is still unclear, many environmental agents have been demonstrated to act as carcinogens in the general population[33]. Of particular importance for carcinogenesis is a class of man-made endocrine disruptors which are widespread environmental contaminants due to their use in plastics manufacturing, including bisphenol A (BPA) and diethylhexyl phthalate (DEHP). Both BPA and DEHP have been associated with carcinogenesis including breast and ovarian tumors[34–37]. While they are thought to primarily act through disrupting the natural hormone signaling leading to hyperproliferation, they have also been found to crosslink to DNA causing adducts[38–47]. Since DNA adducts represent fork blocking lesions and induce mutagenesis, these findings suggest that BPA and DEHP may promote carcinogenesis, at least in part, through genotoxicity. The molecular mechanisms underlying genomic instability induced by these compounds are unknown.

Here, we show that exposure to BPA and DEHP causes accumulation of nascent strand ssDNA gaps during DNA replication. Moreover, we identify the multi-step nucleolytic processing of ssDNA gaps which converts them into cytotoxic double-stranded DNA breaks (DSBs). We show that, unless efficiently repaired by the BRCA pathway, ssDNA gaps are extended bidirectionally by MRE11 in the 3′−5′ direction and by EXO1 in the 5′−3′ direction. Subsequently, the parental strand at the ssDNA gap is cleaved by the MRE11 endonuclease generating a double strand break. Our work sheds light on both the relevance of ssDNA gaps as major determinants of genomic instability, as well as the mechanism through which they are processed to generate genomic instability and cytotoxicity.

## Results

### EXO1 promotes HU and cisplatin-induced ssDNA gaps in BRCA-deficient cells

The MRE11 exonuclease has been previously shown to expand replication-associated ssDNA gaps in the nascent strand[15,27]. MRE11 possess 3′−5′ exonuclease activity, so it conceivably extends the nascent strand ssDNA gap from the 3′ end. We considered what would occur to the 5′ end of the gap. The EXO1 nuclease has 5′−3′ exonuclease activity, and we recently showed that in BRCA-deficient cells, at stalled replication forks, MRE11 and EXO1 cooperate upon fork reversal to extend a ssDNA nick in the nascent strand in both 5′ and 3′ directions[27]. We sought to test if a similar process occurs during nascent strand ssDNA gap processing, where EXO1 employs its 5′−3′ exonuclease activity to extend the gap from the 5′ end (Fig. 1a). We treated HeLa cells with a low-dose (0.4 mM) of hydroxyurea (HU) which was previously shown to induce nascent strand ssDNA gaps[24,27], as opposed to the high-dose (4 mM HU) used to induce degradation of reversed replication forks[1,27]. We then measured ssDNA gap formation using the BrdU alkaline comet assay, as previously employed by us and others[12,48,49]. As we previously described[12,22,27], BRCA2-knockout HeLa cells accumulate replication-associated ssDNA gaps at higher rates than wildtype cells. Depletion of EXO1 suppressed ssDNA gap formation similar to MRE11 knockdown (Fig. 1b and Supplementary Fig. S1a, b), indicating an essential role for EXO1 in ssDNA gap formation/elongation. We also employed the BRCA2^KO EXO1^KO double knockout cells we previously created[27]. In line with the siRNA-mediated depletion experiments, the double knockout cells showed reduced nascent strand ssDNA gap formation compared to the BRCA2^KO single knockout cells (Supplementary Fig. S2).

We next investigated if this activity of EXO1 in gap expansion is restricted to BRCA2-deficient cells or if it also occurs in BRCA1-deficient cells. To test this, we employed the previously described RPE1-BRCA1^KO cells[50]. We noticed that, similar to the BRCA2-deficient cells, the BRCA1-knockout cells also accumulate HU-induced replication-associated ssDNA gaps, which are suppressed upon EXO1 or MRE11 depletion (Fig. 1c). We also generated EXO1-knockout HeLa cells using CRISPR/Cas9. SiRNA-mediated knockdown of BRCA1 or BRCA resulted in increased ssDNA gap accumulation upon treatment with 0.4 mM HU in wildtype HeLa cells, but not in HeLa-EXO1^KO cells (Supplementary Fig. S3a–d). Similarly, inhibition of MRE11 exonuclease activity by the specific inhibitor mirin also suppressed gap formation upon depletion of BRCA1 or BRCA2 in HeLa cells (Supplementary Fig. S3a). Moreover, we extended our analyses to additional DNA damaging agents. It was previously shown that treatment with 150uM cisplatin induces ssDNA gap formation[13,15]. In BrdU alkaline comet assays, we found that cisplatin-induced ssDNA gap formation is also suppressed by depletion of MRE11 or EXO1 (Fig. 1d). Overall, these findings suggest a general role for EXO1 in ssDNA gap expansion.

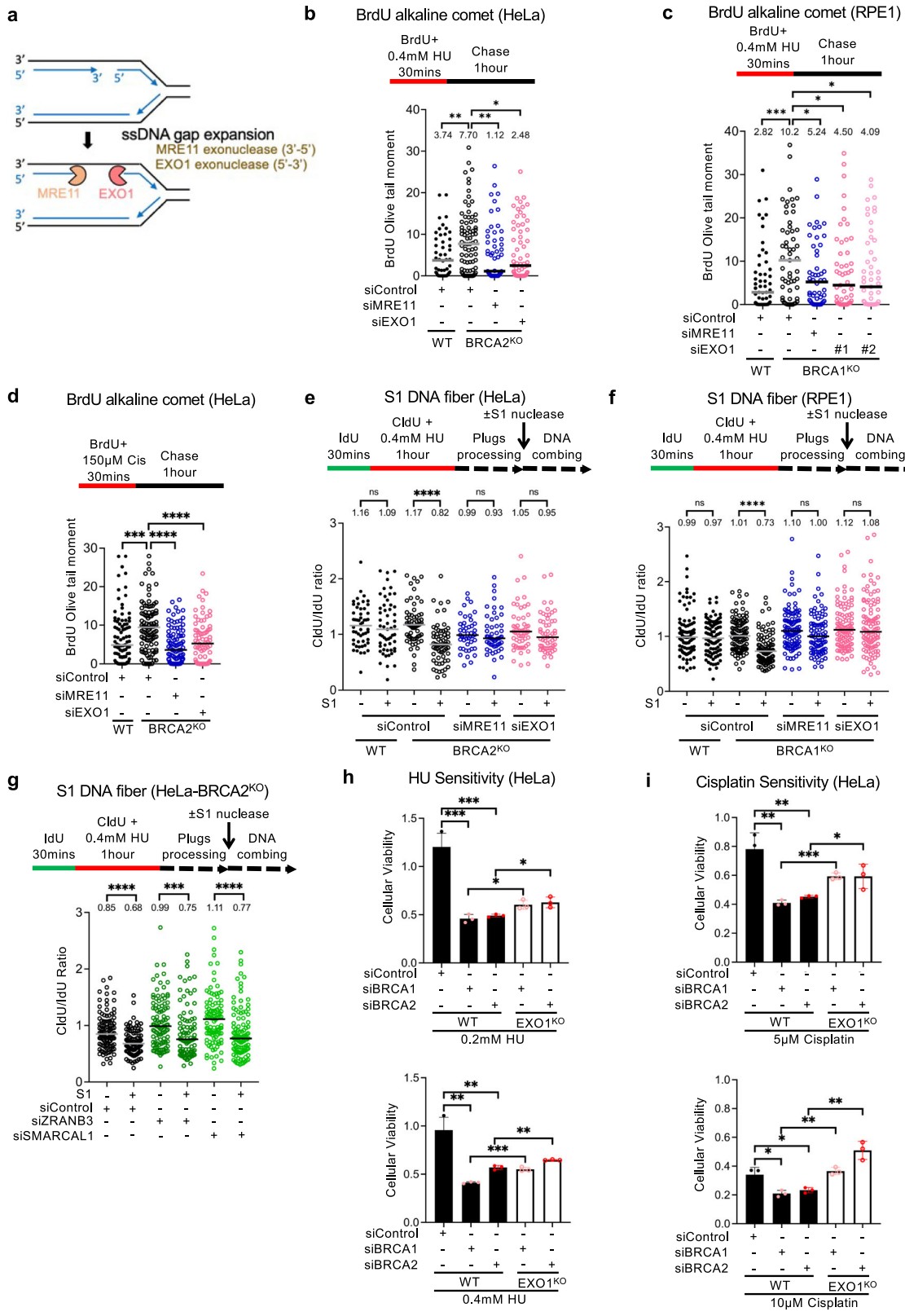

Next, we sought to validate our findings using an independent method to measure nascent strand ssDNA gaps. To this end, we employed the previously-described S1 nuclease DNA fiber combing assay[51]. As expected, in both BRCA2KO (Fig. 1e) and BRCA1KO (Fig. 1f) cells, treatment with 0.4 mM HU resulted in nascent strand ssDNA gaps, as evidenced by the shortening of the CldU/IdU ratio in S1-treated samples compared to untreated samples. In both cell lines, gap

accumulation was suppressed not only by MRE11 depletion, but also by EXO1 depletion. As control, CldU/IdU ratios were not affected by loss of BRCA1, BRCA2, EXO1 or MRE11 in the absence of HU treatment (Supplementary Fig. S4a, b). Moreover, depletion of DNA translocases ZRANB3 and SMARCAL1, responsible for fork reversal[8,9,11], did not affect ssDNA gap accumulation induced by HU treatment in HeLa-BRCA2KO cells (Fig. 1g and Supplementary Fig. S1c, d) arguing that

**Fig. 1 | Loss of EXO1 suppresses the accumulation of nascent strand ssDNA gaps induced by replication stress in BRCA-deficient cells. a** Schematic representation of the ssDNA gap expansion model tested: MRE11 extends the gap from the 3' end and EXO1 extends it from the 5' end. **b, c** BrdU alkaline comet assay showing that EXO1 knockdown suppresses the accumulation of replication-associated ssDNA gaps induced by treatment with 0.4 mM HU in HeLa-BRCA2[KO] (**b**) and RPE1-BRCA1[KO] (**c**) cells, similar to MRE11 depletion. At least 50 nuclei were quantified for each condition. The median values are marked on the graph and listed at the top. Asterisks indicate statistical significance (Mann–Whitney, two-tailed). A schematic representation of the assay conditions is shown at the top. Western blots confirming the EXO1 and MRE11 knockdown are shown in Supplementary Fig. S1a, b. **d** BrdU alkaline comet assay showing that EXO1 knockdown suppresses the accumulation of replication-associated ssDNA gaps induced by treatment with 150 μM cisplatin in HeLa-BRCA2[KO] cells, similar to MRE11 depletion. At least 75 nuclei were quantified for each condition. The median values are marked on the graph and listed at the top. Asterisks indicate statistical significance (Mann–Whitney, two-tailed). A schematic representation of the assay conditions is shown at the top. **e, f** S1 nuclease DNA fiber combing assays showing that knockdown of EXO1 suppresses the accumulation of nascent strand ssDNA gaps induced by treatment with 0.4 mM HU in HeLa-BRCA2[KO] (**e**) and RPE1-BRCA1[KO] (**f**) cells, similar to MRE11 depletion. The ratio of CldU to IdU tract lengths is presented, with the median values marked on the graphs and listed at the top. At least 45 tracts were quantified for each sample. Asterisks indicate statistical significance (Mann–Whitney, two-tailed). Schematic representations of the assay conditions are shown at the top. **g** S1 nuclease DNA fiber combing assays showing that knockdown of ZRANB3 or of SMARCAL1 does not affect the accumulation of nascent strand ssDNA gaps induced by treatment with 0.4 mM HU in HeLa-BRCA2[KO] cells. The ratio of CldU to IdU tract lengths is presented, with the median values marked on the graphs and listed at the top. At least 90 tracts were quantified for each sample. Asterisks indicate statistical significance (Mann–Whitney, two-tailed). Schematic representations of the assay conditions are shown at the top. Western blots confirming the ZRANB3 and SMARCAL1 knockdown are shown in Supplementary Fig. S1c,d. **h, i** Cellular viability assays showing that loss of EXO1 partially suppresses the HU (**h**) and cisplatin (**i**) sensitivity of BRCA1- or BRCA2-knockdown HeLa cells, at two different concentrations as indicated. The average of three independent experiments, with standard deviations indicated as error bars, is shown. Asterisks indicate statistical significance (*t* test, two-tailed, unpaired). Western blots showing EXO1 deletion are shown in Supplementary Fig. S3b. Source data are provided as a Source data file.

MRE11 and EXO1-mediated ssDNA gap formation does not occur on reversed forks. Overall, our results indicate that both the MRE11 3'−5' exonuclease and the EXO1 5'−3' exonuclease are required for ssDNA gap accumulation in BRCA-deficient cells.

Finally, we sought to investigate the impact of ssDNA gap processing by EXO1 on cellular sensitivity. Depletion of BRCA1 or BRCA2 increased the cellular sensitivity to cisplatin and HU, as measured using the CellTiterGlo cellular viability assay (Fig. 1h, i). Deletion of EXO1 partially suppressed this sensitivity in both cases. These findings suggest that ssDNA gap expansion may be partially associated with cellular sensitivity, at least under the circumstances investigated here.

## Direct engagement of MRE11 and EXO1 on ssDNA gaps underlies their role in gap expansion

We next wanted to investigate if the impact of EXO1 and MRE11 on ssDNA gaps described above reflects their direct engagement on gapped DNA, or an indirect effect caused by other putative roles of these nucleases. Previously, the proximity ligation (PLA)-based SIRF (in situ quantification of proteins interactions at DNA replication forks) assay has been employed to detect protein binding to EdU-labeled nascent DNA[52]. In particular, the SIRF assay was used to detect the engagement of MRE11[5] and EXO1[27] on reversed replication forks during fork degradation in BRCA-deficient cells. To induce fork degradation, those experiments were performed in the presence of 4 mM HU. We reasoned that the specific HU treatment conditions may allow us to differentiate between nucleases engagement on reversed forks and their processing of ssDNA gaps. Indeed, treatment with 4 mM HU caused fork degradation in BRCA-deficient cells as investigated by the DNA fiber combing assay (Supplementary Fig. S5a). In contrast, treatment with 0.4 mM HU, the condition we employed above (Fig. 1) to induce nascent strand ssDNA gaps, does not cause fork degradation (Supplementary Fig. S5a). Moreover, as shown above (Fig. 1g), inactivation of fork reversal does not suppress gap formation upon treatment with 0.4 mM HU. Thus, we reasoned that treatment with 0.4 mM HU can be employed to specifically investigate DNA nucleases engagement on ssDNA gaps as opposed of their recruitment to reversed replication forks.

We therefore employed the SIRF assay to measure MRE11 and EXO1 engagement on nascent DNA under these specific conditions (Fig. 2a–f). To measure MRE11 engagement on the 3' end of the gap, we first labeled cells with EdU for 30 min, then washed it away and added 0.4 mM HU for 3 h (Fig. 2a). A similar labeling scheme was previously employed to measure MRE11 engaged in degradation of reversed forks upon treatment of BRCA-deficient cells with 4 mM HU[5,27], so we used this treatment as control. Interestingly, MRE11 was recruited to nascent

DNA in BRCA2-knockout HeLa cells not only upon treatment with 4 mM HU, but also upon treatment with 0.4 mM HU (Fig. 2b). We also found that MRE11 is recruited to nascent DNA in DLD1 cells treated with 0.4 mM HU upon BRCA1 or BRCA2 knockdown (Fig. 2c). Since, as described above, treatment with 0.4 mM HU does not cause fork degradation but instead induces ssDNA gaps, these findings indicate that MRE11 engages nascent DNA at ssDNA gap regions.

We next set out to investigate if EXO1 is also recruited to nascent strand gaps. The EXO1 SIRF signal was low when assayed using the labeling scheme described above for MRE11, which labels the 3' end of the gap (Supplementary Fig. S6a). Since EXO1 exonuclease activity operates in the opposite direction than that of MRE11, we sought to instead label the 5' end of the gap, which we hypothesized to be engaged by EXO1. To achieve this, we treated cells with 0.4 mM HU for 30 mins, followed by another 30 mins during which we also added EdU (Fig. 2d). Using these experimental conditions, we found that EXO1 is specifically recruited to nascent DNA in BRCA2-knockout HeLa cells (Fig. 2e). MRE11 SIRF signal was also detected under these conditions, but this is not unexpected since the 3' gap ends may also be labeled using this scheme (Supplementary Fig. S6b, c). Next, we measured EXO1 recruitment to nascent strands, using a similar setup, in DLD1 cells. Since these cells showed reduced EdU incorporation upon HU treatment, we altered the labeling scheme to incubate with EdU and 0.4 mM HU for only 20 min, without pre-incubation with HU. Under these conditions, we observed increased nascent strand recruitment of EXO1 upon depletion of BRCA1 or BRCA2 (Fig. 2f). Moreover, depletion of the fork reversal translocase ZRANB3 did not affect either MRE11 or EXO1 recruitment to nascent DNA as analyzed using the respective labeling schemes (Fig. 2g, h), ruling out that the SIRF signal detected under these conditions arises from binding of these nucleases to reversed forks. Finally, depletion of the primase-polymerase PRIMPOL responsible for gap formation reduced EXO1 recruitment (Fig. 2h and Supplementary Fig. S1e), further confirming that our experimental conditions detect EXO1 binding to ssDNA gaps. Overall, these findings suggest that EXO1 engages nascent DNA at ssDNA gaps region from their 5' end.

## Exposure to environmental carcinogens BPA and DEHP causes nascent strand ssDNA gaps

While ssDNA gap accumulation has been associated with chemosensitivity, arguing that it promotes cytotoxic DNA damage, this has been mostly investigated in specific genetic backgrounds (BRCA deficiency or PRIMPOL overexpression) upon treatment with 0.4 mM HU as a general condition of replication stress, or upon treatment with genotoxic chemotherapeutic agents such as cisplatin or PARP

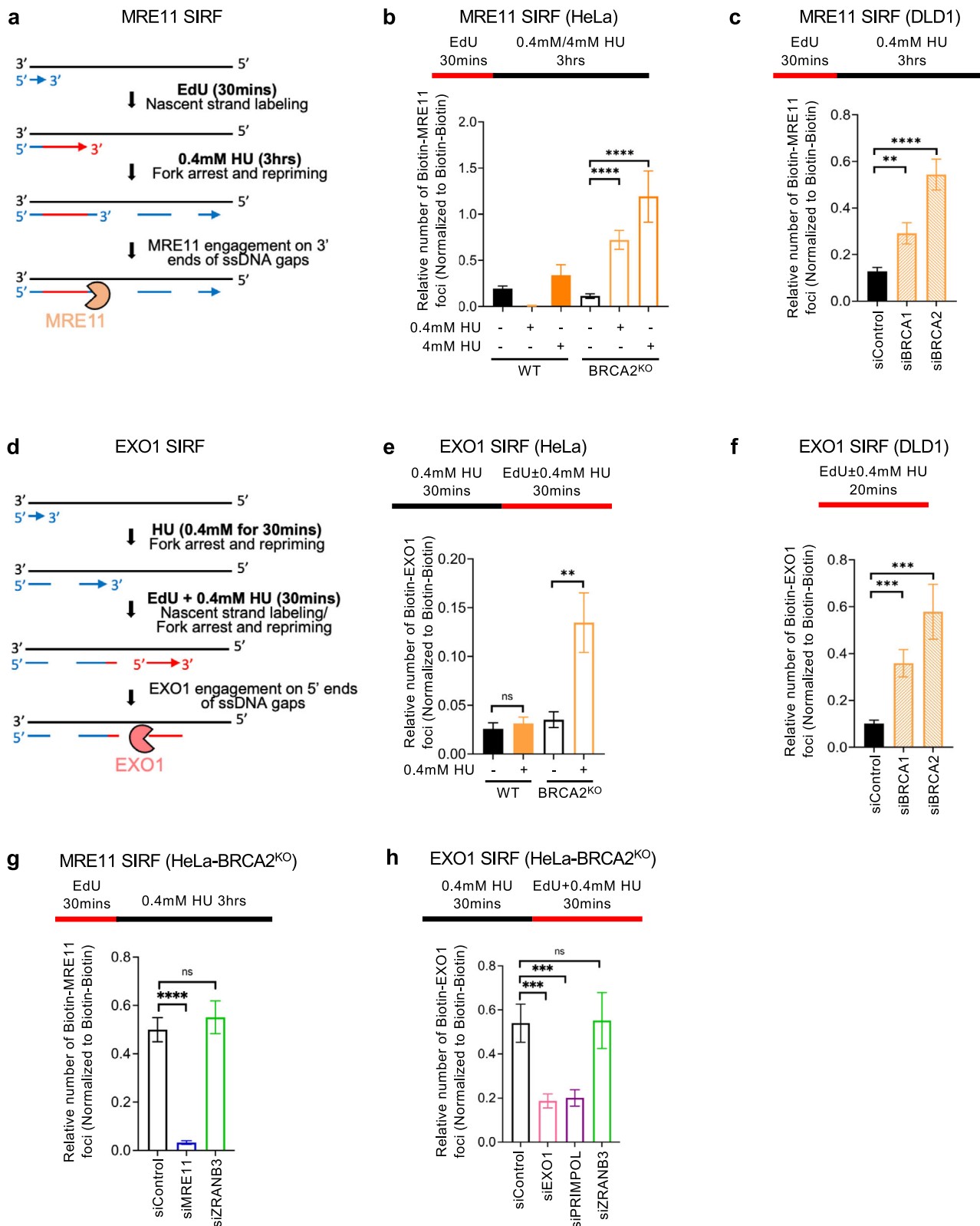

inhibitors[13,15–19,22–26]. We sought to investigate if ssDNA gap formation is specific to these processes, or if instead it represents a general mechanism of DNA damage hypersensitization and genomic instability. In particular, we wanted to explore the relevance of ssDNA gaps not only for cancer therapy, but also for initial carcinogenesis. To this end, we employed two potentially carcinogenic endocrine disruptors which are ubiquitous man-made environmental contaminants employed in

plastics manufacturing, namely bisphenol A (BPA) and diethylhexyl phthalate (DEHP). Both compounds are associated with breast and ovarian carcinogenesis[34–37] and have been shown to cause DNA adducts and mutagenesis[38–47], suggesting that they may promote carcinogenesis, at least in part, through genotoxicity. To address this, we first employed the neutral comet assay and γH2AX foci detection by immunofluorescence to investigate if treatment with BPA or DEHP

**Fig. 2 | MRE11 and EXO1 localize to replication stress-induced nascent strand ssDNA gaps. a–c** SIRF experiments showing that treatment with 0.4 mM HU induces binding of MRE11 to nascent DNA in HeLa-BRCA2^KO cells (**b**) or upon depletion of BRCA1 or BRCA2 in DLD1 cells (**c**), similar to treatment with 4 mM HU. The labeling scheme (**a**) is designed to capture MRE11 binding to the 3′ end of the gap (for simplicity, only one strand, e.g., the leading strand, is shown in the schematic representation; EdU-labeled nascent DNA is indicated in red). At least 50 cells were quantified for each condition. Bars indicate the mean values, error bars represent standard errors of the mean, and asterisks indicate statistical significance (*t* test, two-tailed, unpaired). Schematic representations of the assay conditions are shown at the top. Western blots confirming the BRCA1 and BRCA2 knockdown are shown in Supplementary Fig. S3c, d. **d**, **e** SIRF experiments showing that treatment with 0.4 mM HU induces binding of EXO1 to nascent DNA in HeLa-BRCA2^KO cells. The labeling scheme (**d**) is designed to capture EXO1 binding to the 5′ end of the gap (for simplicity, only one strand, e.g., the leading strand, is shown in the schematic representation; EdU-labeled nascent DNA is indicated in red). At least 60 cells were quantified for each condition. Bars indicate the mean values, error bars represent standard errors of the mean, and asterisks indicate statistical significance (*t* test, two-tailed, unpaired). Schematic representations of the assay conditions are shown at the top. **f** SIRF experiments showing that treatment with 0.4 mM HU induces binding of EXO1 to nascent DNA upon depletion of BRCA1 or BRCA2 in DLD1 cells. At least 60 cells were quantified for each condition. Bars indicate the mean values, error bars represent standard errors of the mean, and asterisks indicate statistical significance (*t* test, two-tailed, unpaired). Schematic representations of the assay conditions are shown at the top. **g**, **h** SIRF experiments showing that ZRANB3 knockdown does not affect binding of MRE11 (**g**) or EXO1 (**h**) induced by treatment of HeLa-BRCA2^KO cells with 0.4 mM HU. At least 80 cells were quantified for each condition. Depletion of MRE11 or EXO1 respectively is used as control to confirm the specificity of the SIRF signals observed. Bars indicate the mean values, error bars represent standard errors of the mean, and asterisks indicate statistical significance (*t* test, two-tailed, unpaired). Schematic representations of the assay conditions are shown at the top. Source data are provided as a Source data file.

causes DSB accumulation (Fig. 3a–f). We exposed cells to 200 μM BPA or 200 μM DEHP for 2 h, treatment conditions which are within the same range as previously employed in cell culture studies[38–40,44,53]. In multiple cell lines, including HeLa (Fig. 3a, d), RPE1 (Fig. 3b, e), and DLD1 (Fig. 3c, f) we observed increased DSB formation as measured by both the neutral comet assay (Fig. 3a–c) and the γH2AX immunofluorescence assay (Fig. 3d–f). While the DSB formation was evident in BRCA1/2-deficient cells, we also observed DSB induction by these compounds in BRCA-wildtype cells, especially in the case of BPA (at the concentrations investigated here). These findings indicate that both BPA and DEHP are genotoxic.

In order to understand how treatment with these agents causes DSBs, we first sought to investigate if they cause fork degradation in the DNA fiber combing assay. Using a labeling scheme employed to measure degradation of reversed replication forks (Supplementary Fig. S5a), we found that, in both wildtype and BRCA-deficient cells, treatment with 200 μM BPA did not change the CldU/IdU ratio from 1 (Fig. 3g and Supplementary Fig. S5b, c), arguing that BPA treatment does not induce degradation of stalled replication forks.

Because of the growing relevance of ssDNA gap metabolism for genomic stability, we next explored if gap formation underlies the genotoxicities of BPA and DEHP. To this end, we measured nascent strand ssDNA gap formation upon treatment with these agents under the same conditions (200 μM for 2 h) used to measure DSB formation as described above. BrdU alkaline comet assays revealed that both BPA and DEHP cause replication-associated ssDNA gaps in BRCA2-knockout HeLa cells (Fig. 3h) and BRCA1-knockout RPE1 cells (Fig. 3i). BPA and DEHP increased gap formation not only in BRCA-deficient cells, but also in BRCA-wildtype HeLa cells.

We next validated these findings using the S1 nuclease DNA fiber combing assay. Similar to the results using the BrdU alkaline comet assay, treatment with 200 μM BPA or DEHP caused nascent strand ssDNA gaps in both HeLa and RPE1 cells (Fig. 3j–l). While gap formation was more prominent in BRCA1/2-knockout cells as evidenced by a lower CldU/IdU ratio in S1-treated samples compared to non-treated samples, both compounds induced gaps in wildtype cells as well, in both HeLa (Fig. 3j) and RPE1 (Fig. 3k) cells. In contrast, treatment with 0.4 mM HU induced nascent strand gaps in BRCA-deficient, but not in wildtype HeLa cells (Fig. 3l), in line with previous reports[15,24]. These findings suggest that exposure of normal (BRCA-proficient) cells to potential carcinogens BPA and DEHP, perhaps upon adduct formation, causes nascent strand ssDNA gap formation, which is exacerbated by BRCA deficiency.

### Nascent strand ssDNA gaps induced by BPA and DEHP are processed by MRE11 and EXO1 exonucleases

To further understand the relevance of BPA and DEHP-induced ssDNA gap formation, we sought to identify the mechanism of their processing.

Since our results reported above (Fig. 2) using 0.4 mM HU to induce nascent strand gaps indicated their bidirectional processing by MRE11 and EXO1 nucleases, we sought to determine if a similar processing occurs for ssDNA gaps induced by BPA and DEHP. We first employed the BrdU alkaline comet assay to measure the impact of MRE11 and EXO1 on replication-associated ssDNA gap induction by BPA and DEHP in HeLa cells. EXO1 deletion (Fig. 4a) or depletion (Fig. 4b) suppressed BPA and DEHP-induced gap formation. Similarly, MRE11 depletion (Fig. 4b) or inhibition of its exonuclease activity by mirin (Fig. 4c) suppressed ssDNA gaps induced by BPA and DEHP. We next validated these findings using the S1 nuclease DNA fiber combing assay. In both HeLa-BRCA2^KO (Fig. 4d) and RPE1-BRCA1^KO (Fig. 4e) cells, depletion of MRE11 or of EXO1 suppressed nascent strand ssDNA accumulation. As was also the case for ssDNA gaps induced by treatment with 0.4 mM HU, depletion of fork reversal translocases ZRANB3 and SMARCAL1 did not affect ssDNA gap accumulation upon BPA treatment (Fig. 4f), arguing that these gaps do not form on reversed forks.

Finally, we employed the SIRF assay to investigate the dynamics of MRE11 and EXO1 recruitment to nascent DNA upon BPA exposure, using the labeling schemes described above (Fig. 2) to measure the engagement of MRE11 and EXO1 to gaps induced by treatment with 0.4 mM HU. Treatment of HeLa-BRCA2^KO cells with 200 μM BPA, which, as described above induces nascent strand gaps but not fork degradation (Fig. 3), resulted in recruitment of MRE11 to nascent DNA similarly to treatment with 0.4 mM HU (Fig. 4g). Similar findings were observed upon BRCA1 or BRCA2 depletion in DLD1 cells (Fig. 4h, i). MRE11 recruitment upon BPA treatment was not affected by loss of ZRANB3 (Fig. 4j), indicating that it does not occur on reversed forks. Finally, EXO1 binding to the 5′ end of ssDNA gaps was also observed upon treatment with 200 μM BPA in BRCA-deficient HeLa (Fig. 4k) and DLD1 cells (Fig. 4l, m), and was also not dependent on ZRANB3 (Fig. 4n). These results indicate that, similar to HU-induced ssDNA gaps, nascent strand gaps induced by BPA are engaged by MRE11 and EXO1 for bidirectional gap expansion.

### Regulation of EXO1 and MRE11 recruitment to ssDNA gaps

In both the BrdU alkaline comet and the S1 nuclease DNA fiber combing assays, loss of MRE11 and EXO1 resulted in an apparent nearly-complete suppression of ssDNA gap accumulation. Indeed, when we found that, both with 0.4 mM HU and 200 μM BPA, treatment of EXO1-depleted HeLa-BRCA2^KO cells with mirin did not further suppress ssDNA gap formation (Fig. 5a, b), suggesting an epistatic relationship between EXO1 and MRE11 under these conditions. To further explore this, we investigated the recruitment of EXO1 and MRE11 to nascent DNA under gap-forming conditions. Using PLA assays, we found that EXO1 and MRE11 co-localize upon treatment with 0.4 mM HU or 200 μM BPA (Fig. 5c, d). These findings suggest that EXO1 and MRE11 may engage ssDNA gaps as a complex.

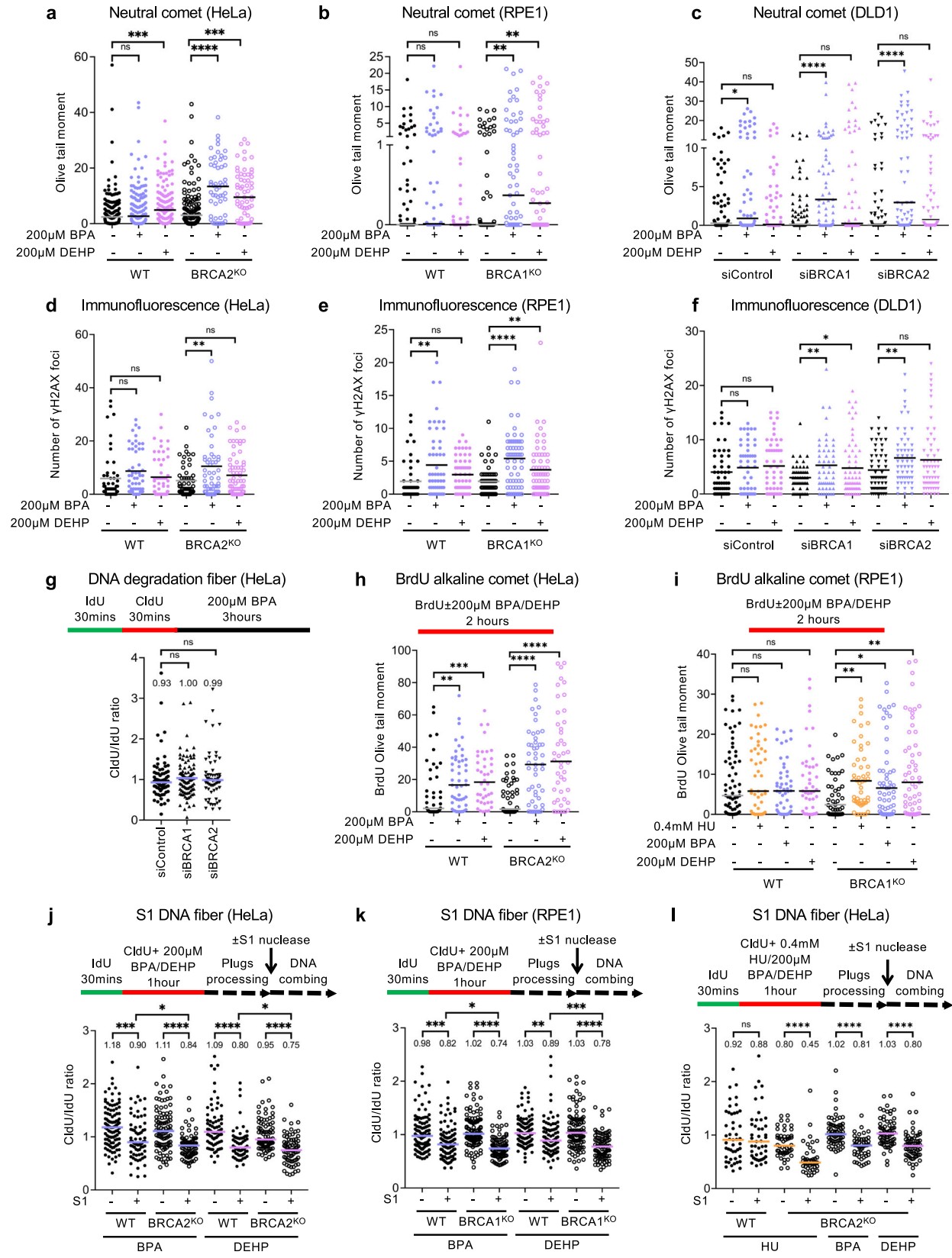

Next, we investigated if recruitment of MRE11 and EXO1 to ssDNA gaps is co-dependent. Loss of MRE11 suppressed EXO1 binding to nascent DNA upon 0.4 mM HU treatment, as measured using the SIRF assay (Fig. 5e). In contrast, EXO1 depletion did not affect MRE11 binding to nascent DNA under these conditions (Fig. 5f). These suggest that

EXO1 engagement on ssDNA gaps requires MRE11, while MRE11 can also be recruited in the absence of EXO1. Overall, our studies suggest that EXO1 and MRE11 engage ssDNA in a co-regulated manner.

Surprisingly, inhibition of MRE11 endonuclease activity using the specific inhibitor PFM01 partially suppressed the ssDNA gaps induced

**Fig. 3 | BPA and DEHP cause nascent strand ssDNA gaps. a–c** Neutral comet assays showing that treatment with 200 μM BPA or 200 μM DEHP for 2 h causes DSBs in HeLa (**a**), RPE1 (**b**), and DLD1 (**c**) cells, particularly upon BRCA1/2 knockout (**a, b**) or knockdown (**c**). At least 50 comets were quantified for each sample. The median values are marked on the graph, and asterisks indicate statistical significance (Mann–Whitney, two-tailed). **d–f** γH2AX immunofluorescence showing that that treatment with 200 μM BPA or 200 μM DEHP for 2 h increases γH2AX foci in HeLa (**d**), RPE1 (**e**), and DLD1 (**f**) cells, particularly upon BRCA1/2 knockout (**d, e**) or knockdown (**f**). At least 50 cells were quantified for each condition. The mean value is represented on the graphs, and asterisks indicate statistical significance (*t* test two-tailed, unpaired). **g** DNA fiber combing assay in HeLa cells showing that treatment with 200 μM BPA does not cause fork degradation. The ratio of CldU to IdU tract lengths is presented, with the median values marked on the graphs and listed at the top. At least 60 tracts were quantified for each sample. Asterisks indicate statistical significance (Mann–Whitney, two-tailed). Schematic

representations of the assay conditions are shown at the top. **h, i** BrdU alkaline comet assay showing that treatment with 200 μM BPA or 200 μM DEHP for 2 h results in accumulation of replication-associated ssDNA gaps in HeLa wildtype and BRCA2-knockout cells (**h**), as well as in BRCA1-knockout RPE1 cells (**i**). At least 40 nuclei were quantified for each condition. The median values are marked on the graph. Asterisks indicate statistical significance (Mann–Whitney, two-tailed). **j–l** S1 nuclease DNA fiber combing assays showing that that treatment with 200 μM BPA or 200 μM DEHP results in accumulation of nascent strand ssDNA gaps in HeLa (**j, l**) and RPE1 (**k**) cells. Gap accumulation is more pronounced in BRCA2-knockout (**j, l**) and BRCA1-knockout (**k**) cells compared to the respective wildtype controls. In contrast, treatment with 0.4 mM HU causes gaps only in BRCA2-knockout, but not in wildtype cells (**l**). At least 40 tracts were quantified for each sample. Asterisks indicate statistical significance (Mann–Whitney, two-tailed). Schematic representations of the assay conditions are shown at the top. Source data are provided as a Source data file.

by both HU and BPA in the BrdU alkaline assay (Supplementary Fig. S7a, b), perhaps suggesting that MRE11 endonuclease may also engage dsDNA directly and create nicks on nascent DNA. Perhaps in line with this, MRE11 SIRF experiments showed that, unlike the case for EXO1, PRIMPOL depletion does not suppress MRE11 recruitment to nascent DNA upon treatment with 0.4 mM HU (Supplementary Fig. S7c).

## ssDNA gaps bidirectionally extended by EXO1 and MRE11 exonuclease activities are converted into DSBs by the MRE11 endonuclease activity

How ssDNA gaps cause genomic instability and cytotoxicity is unclear and a matter of debate. One model proposed that DSBs arise as a consequence of replicating through the ssDNA gap in the subsequent cell cycle[26]. Another model hypothesized that DSBs are induced subsequently during apoptosis and not directly formed from the ssDNA gaps[24]. A recent report identified a role for the POLQ polymerase in ssDNA gap filling and showed that in POLQ-deficient cells replication forks are processed into double strand breaks, which are suppressed by inhibiting the endonuclease activity of MRE11[17]. In our own hands, we found that treatment of HeLa-BRCA2[KO] cells with 0.4 mM HU results in accumulation of DSBs as measured by the neutral comet assay, which was suppressed by depletion of the primase-polymerase PRIMPOL responsible for gap formation (Fig. 6a). These findings, together with our findings reported above showing the involvement of MRE11 and EXO1 nucleases in ssDNA gap processing (Figs. 2 and 4), suggest a stepwise model of sequential engagement of nucleases in nascent strand ssDNA gap processing, with the MRE11 and EXO1 exonuclease activities extending a small ssDNA gap bidirectionally from the 3' and 5' ends respectively. This is followed by the MRE11 endonuclease activity cleaving the parental ssDNA strand at the gap region to result in a DSB. To test this model, we measured the impact of ssDNA gap processing factors on DSB formation using the neutral comet and γH2AX foci immunofluorescence assays.

We first investigated the role of MRE11 endonuclease activity in converting ssDNA gaps into DSBs. Neutral comet assays showed that treatment of HeLa-BRCA2[KO] cells with 0.4 mM HU or 200 μM BPA for 2 h resulted in DSB accumulation (Fig. 6b). Under these treatment conditions, we could not detect cleaved Caspase-3 by western blot, or induction of apoptosis as measured by Annexin V flow cytometry (Supplementary Fig. S8a, b), indicating that these DSBs are not formed as a result of apoptosis initiation. In line with this, treatment with the apoptosis inhibitor Z-VAD-FMK did not affect DSB accumulation under these conditions (Supplementary Fig. S8c). In contrast, treatment with the MRE11 endonuclease inhibitor PFM01 suppressed DSB formation induced by either 0.4 mM HU or 200 μM BPA (Fig. 6b). Since these treatment conditions induce ssDNA gaps but not fork degradation (Figs. 1–3), these findings suggests that MRE11 endonuclease activity cleaves the template strand to convert a ssDNA gap into a DSB. Next,

we investigated the impact of exonuclease activities on the generation of DSBs from ssDNA gaps. MRE11 exonuclease inhibition by mirin suppressed DSB induction upon treatment with 0.4 mM HU or 200 μM BPA to a similar extend as its endonuclease inhibition by PFM01 (Fig. 6b). Moreover, EXO1 knockout (Fig. 6b) or knockdown (Fig. 6c) similarly suppressed DSB formation under these conditions. These findings suggest that endonucleolytic processing by MRE11 to generate DSBs cannot occur without prior exonucleolytic gap expansion by MRE11 and EXO1.

We next confirmed these results using γH2AX foci as a readout of DSB formation (Fig. 6d–f). Similar to the neutral comet results described above, inhibition of MRE11 endonuclease activity (by PFM01 treatment), inhibition of MRE11 exonuclease activity (by mirin treatment), and depletion or deletion of EXO1 exonuclease all resulted in suppression of γH2AX foci induced by treatment with 0.4 mM HU (Fig. 6d, e) or 200 μM BPA (Fig. 6f). Finally, we measured the impact of the expansion of BPA-induced ssDNA gaps on cellular sensitivity. Knockdown of BRCA1 or BRCA2 depletion causes increased cellular sensitivity to BPA (Fig. 6g). This increase was suppressed by deletion of EXO1. These findings argue that expansion of BPA-induced ssDNA gaps may contribute to the cellular sensitivity to this agent.

Altogether, our findings show that MRE11 and EXO1 exonuclease activities promote the conversion of ssDNA gaps into DSBs and indicate a stepwise processing of replication-dependent ssDNA gaps in which MRE11 and EXO1 exonucleases bidirectionally expand a short ssDNA gap in the nascent strand, while MRE11 endonuclease activity subsequently cleaves the intact template ssDNA strand to create a DSB (Fig. 7).

## Discussion

Our work sheds new light on the mechanism of ssDNA gap processing in BRCA-deficient cells. In particular, we (1) identify a role for EXO1 in nascent strand ssDNA gap expansion, and (2) identify the MRE11 endonuclease activity as transforming the ssDNA gaps into dsDNA breaks.

It was recently shown that MRE11 exonuclease inhibition suppresses nascent strand gap formation in BRCA-deficient cells[15,27], suggesting that it extends unfilled gaps in the 3'–5' direction. Whether these gaps are also processed from their 5' end was not known. We show here that the 5'–3' exonuclease activity of EXO1 extends the gap from the 5' ends, and this is an essential step in the conversion of ssDNA gaps into genotoxic lesions. This model is in line with previous reports that EXO1 depletion reduces the accumulation of ssDNA at replication forks upon exposure to agents that cause formation of bulky DNA adducts, such as BPDE[54] and UV[55], suggesting a general role of EXO1 in processing the nascent strand at stalled replication forks.

Upon EXO1 depletion in BRCA-deficient cells, we observed a suppression of ssDNA gap length in the S1 nuclease DNA fiber combing assay, by measuring the CldU/IdU ratio. This is somewhat puzzling, as

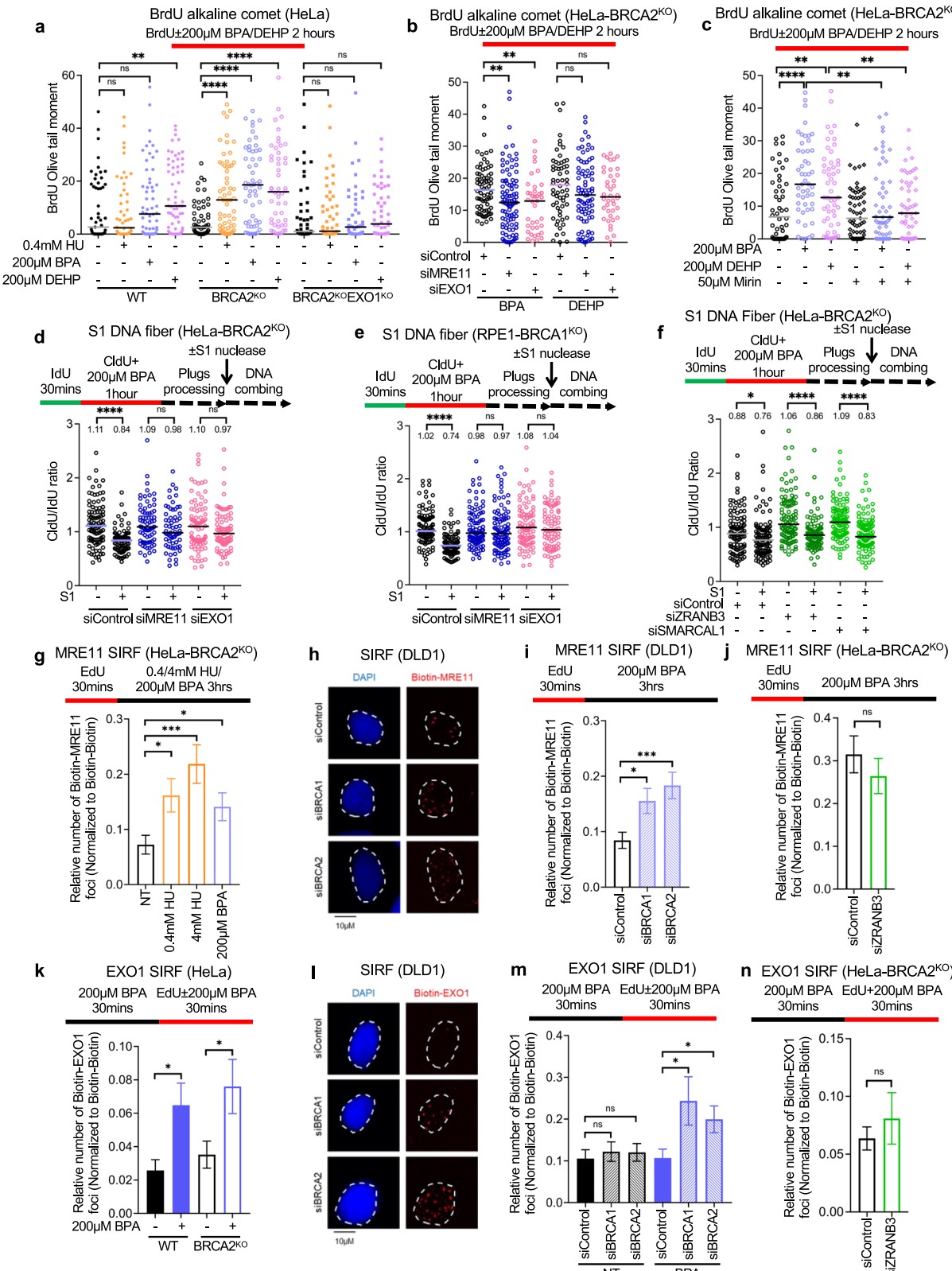

EXO1 expands the gap from the 5′ end, moving away from the labeled tract, and thus not shortening the distance from the 3′ end of the gap to the IdU-to-CldU transition point. One explanation for this result is that, by expanding the gap, even if this is in the other direction, the activity of EXO1 generates a longer stretch of ssDNA in the template strand, thus providing more opportunities for the S1 nuclease to cleave the nascent strand, ultimately resulting in shorter CldU/IdU ratios. On

the other hand, this finding is in line with the epistasis between MRE11 and EXO1 we observed when measuring ssDNA gaps using the BrdU alkaline comet assay, and may reflect a co-regulation between MRE11 and EXO1 activities. Indeed, we show that MRE11 and EXO1 co-localize under ssDNA gap-inducing conditions, and that EXO1 engagement on ssDNA gaps depends on MRE11. In contrast, we found that MRE11 recruitment to nascent DNA upon 0.4 mM HU treatment does not

**Fig. 4 | BPA-induced nascent strand gaps are extended by MRE11 and EXO1.**
**a**–**c** BrdU alkaline comet assay showing that EXO1 knockout (**a**) or knockdown (**b**), or MRE11 knockdown (**b**) or inhibition (**c**) suppresses the accumulation of replication-associated ssDNA gaps induced by treatment with 200 μM BPA or 200 μM DEHP in HeLa wildtype and BRCA2-knockout cells. At least 39 nuclei were quantified for each condition. The median values are marked on the graph. Asterisks indicate statistical significance (Mann–Whitney, two-tailed). **d, e** S1 nuclease DNA fiber combing assays showing that knockdown of MRE11 or of EXO1 suppresses the accumulation of nascent strand ssDNA gaps induced by treatment with 200 μM BPA in HeLa-BRCA2$^{KO}$ (**d**) and RPE1-BRCA1$^{KO}$ (**e**) cells. The ratio of CldU to IdU tract lengths is presented, with the median values marked on the graphs and listed at the top. At least 65 tracts were quantified for each sample. Asterisks indicate statistical significance (Mann–Whitney, two-tailed). Schematic representations of the assay conditions are shown at the top. **f** S1 nuclease DNA fiber combing assays showing that knockdown of ZRANB3 or of SMARCAL1 does not affect the accumulation of nascent strand ssDNA gaps induced by treatment with 200 μM BPA in HeLa-BRCA2$^{KO}$ cells. The ratio of CldU to IdU tract lengths is presented, with the median values marked on the graphs and listed at the top. At least 85 tracts were quantified for each sample. Asterisks indicate statistical significance (Mann–Whitney, two-tailed). Schematic representations of the assay conditions are shown at the top. **g**–**i** SIRF experiments showing that treatment with 200 μM BPA induces binding of MRE11 to nascent DNA in BRCA-deficient HeLa (**g**) or DLD1 (**h, i**) cells, similar to treatment with 4 mM HU. The labeling scheme is designed to capture MRE11 binding to the 3′ end of the gap. Representative

micrographs, with scale bars representing 10 μm (**h**) and quantifications (**g, i**) are shown. At least 70 cells were quantified for each condition. Bars indicate the mean values, error bars represent standard errors of the mean, and asterisks indicate statistical significance (*t* test, two-tailed, unpaired). Schematic representations of the assay conditions are shown at the top. **j** SIRF experiments showing that MRE11 binding to nascent DNA upon treatment with 200 μM BPA in HeLa-BRCA2$^{KO}$ cells is not affected by ZRANB3 depletion. At least 68 cells were quantified for each condition. Bars indicate the mean values, error bars represent standard errors of the mean, and asterisks indicate statistical significance (*t* test, two-tailed, unpaired). Schematic representations of the assay conditions are shown at the top. **k**–**m** SIRF experiments showing that treatment with 200 μM BPA induces binding of EXO1 to nascent DNA in HeLa-BRCA2$^{KO}$ cells (**k**) or upon depletion of BRCA1 or BRCA2 in DLD1 cells (**l, m**). The labeling scheme is designed to capture EXO1 binding to the 5′ end of the gap. Representative micrographs, with scale bars representing 10 μm (**l**) and quantifications (**k, m**) are shown. At least 60 cells were quantified for each condition. Bars indicate the mean values, error bars represent standard errors of the mean, and asterisks indicate statistical significance (*t* test, two-tailed, unpaired). Schematic representations of the assay conditions are shown at the top. **n** SIRF experiments showing that EXO1 binding to nascent DNA upon treatment with 200 μM BPA in HeLa-BRCA2$^{KO}$ cells is not affected by ZRANB3 depletion. At least 80 cells were quantified for each condition. Bars indicate the mean values, error bars represent standard errors of the mean, and asterisks indicate statistical significance (*t* test, two-tailed, unpaired). Schematic representations of the assay conditions are shown at the top. Source data are provided as a Source data file.

depend on depend on EXO1, suggesting that MRE11 can be recruited to ssDNA gaps independently of EXO1, but cannot exert its gap expansion activity in the absence of EXO1. Alternatively, this finding may indicate that MRE11 also binds to nascent DNA at other structures than ssDNA gaps, since its recruitment is not suppressed by PRIMPOL depletion. Indeed, the finding that MRE11 endonuclease inhibition can partially suppress ssDNA gap formation suggests that MRE11 can also initiate ssDNA gaps independently of PRIMPOL, through its endonuclease activity.

A limitation of our study is that we cannot formally rule out that some of the ssDNA gaps observed occur on reversed forks, considering the multiple fork reversal activities present in cells. In addition, we cannot rule out that the DSBs generated by nuclease activities under gap-inducing conditions are also formed from other structures, such as stalled replication forks. Nevertheless, our findings that PRIMPOL depletion suppresses DSB formation induced by treatment with 0.4 mM HU in BRCA-deficient cells suggests that ssDNA gaps represent a significant source of DSBs in these cells. Our work suggests that, unless efficiently filled, ssDNA gaps are subjected to a multi-step process of nucleolytic conversion, with MRE11 and EXO1 exonucleases expanding the gap bidirectionally from the 3′ and 5′ ends respectively, followed by endonucleolytic cleavage of the template strand at the gap region to form a DSB (Fig. 7a). The coordination of MRE11 and EXO1 endonuclease/exonuclease activities to process DNA structures has been previously described in other contexts. We previously showed that, in BRCA-deficient cells, the nascent strands on the reversed arm at stalled replication forks are initially cleaved by the endonuclease activity of MRE11 in the proximity of the DSB end, forming a ssDNA nick which allows bidirectional engagement of MRE11 exonuclease activity in the 3′−5′ direction towards the DSB end, and of EXO1 exonuclease in the 5′−3′ direction for long range resection towards the fork junction and beyond[27] (Fig. 7b). A similar processing mechanism had been previously shown to take place during DNA end processing at DSB sites undergoing HR-mediated repair[56–65]. An MRE11 endonuclease cleavage in the proximity of the DSB end allows engagement of MRE11 exonuclease towards the break and engagement of EXO1 exonuclease to catalyze long-range resection in the opposite direction, moving away from the break. This results in formation of a long 5′ overhang that can invade the homologous sister chromatid (Fig. 7c). In all three cases, the MRE11 3′−5′ exonuclease activity and the EXO1 5′−3′ exonuclease activity bidirectionally expand a ssDNA nick or short gap. In ssDNA gap

processing (Fig. 7a), the MRE11 endonuclease cleaves the template strand to cause a DSB after gap expansion by exonucleases. In contrast, during fork degradation (Fig. 7b) and DSB end resection (Fig. 7c), the endonuclease activity of MRE11 is engaged first, since in both these cases DSB ends are protected by binding of the KU complex, which prevents the direct engagement of exonuclease activities. The endonuclease activity of MRE11, which is recruited to dsDNA in the proximity of the KU-bound DSB end, is required to create the initial nick extended by the exonuclease activities or EXO1 and MRE11. MRE11-mediated exonucleolytic degradation towards the DSB end results in removal of the KU complex, while EXO1 catalyzes long-range resection in the opposite direction (moving away from the DSB end).

Our work sheds additional light on the ongoing debate regarding the mechanism of cytotoxicity of ssDNA gaps. Previous models proposed that ssDNA gaps are converted to DSBs during DNA replication in the subsequent cell cycle or alternatively that DSBs are only generated unspecifically, during apoptosis-mediated cell death[21,24,26,30]. Our results suggest that, at least in part, DSBs arise from the local and immediate activity of MRE11 endonuclease on the template strand at the ssDNA gap region upon bidirectional ssDNA gap expansion by the opposing exonucleolytic activities of MRE11 and EXO1 (Fig. 6a). This is, in fact, in line with previous findings in *Xenopus* egg extracts showing that, in the absence of POLQ-mediated gap filling, MRE11 endonuclease activity causes replication fork breakage[17]. Interestingly, it was previously shown that BRCA-mutant but HR-proficient cells are still chemosensitive if they accumulate ssDNA gaps[24]. This raises the question of why DSBs generated from ssDNA gaps are not repaired through HR but instead are cytotoxic. We speculate that, unlike canonical DSBs (such as those generated by ionizing radiation, DSB cutting enzyme such as the SceI and Cas9 nucleases, or other agents that directly disrupt the two DNA strands), DSBs generated from ssDNA gaps are not repairable by HR since they have long resected ends of different polarity than those needed for HR (Supplementary Fig. S9). In order for such a break to be repairable through HR, the 5′ end of the break on the template strand would need to be resected farther beyond the complementary nascent strand end, to reveal a 3′ overhang. This is unlikely to occur if the nascent strand is continuously resected by MRE11 and EXO1, and thus the second end capture cannot take place. Instead, we speculate that the only way a DSB generated through ssDNA gap processing can be fixed is through break-induced replication (BIR), a highly mutagenic process. This model is potentially in line with the

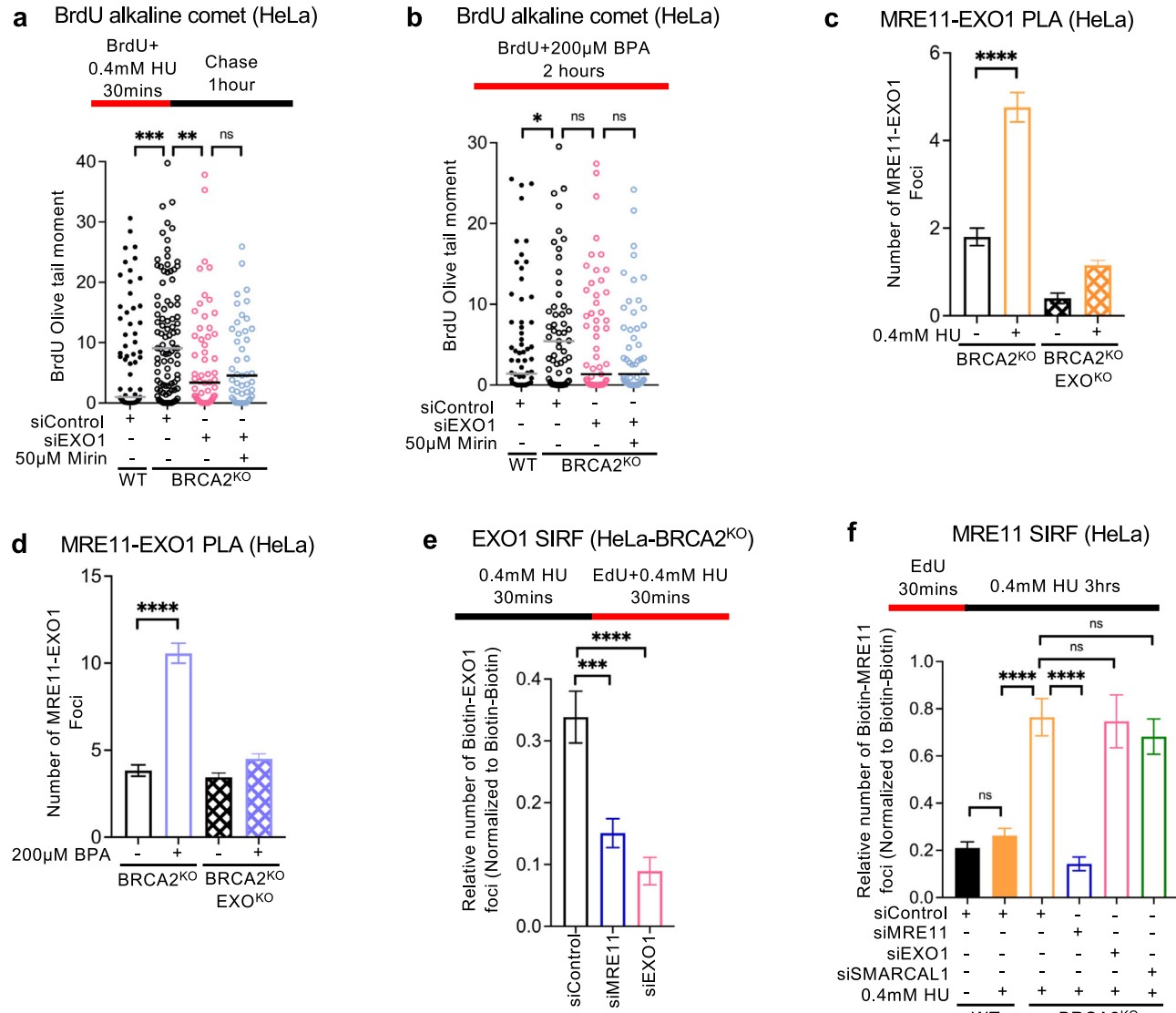

**Fig. 5 | Regulation of EXO1 and MRE11 recruitment to nascent strand ssDNA gaps. a, b** BrdU alkaline comet assays showing that treatment of EXO1-depleted HeLa-BRCA2[KO] cells with mirin does not further suppress the accumulation of ssDNA gaps induced by HU (**a**) or BPA (**b**) exposure. At least 40 nuclei were quantified for each condition. The median values are marked on the graph. Asterisks indicate statistical significance (Mann–Whitney, two-tailed). **c, d** PLA assays showing that EXO1 and MRE11 co-localize upon treatment with 0.4 mM HU (**c**) or 200 μM BPA (**d**) for 3 h. EXO1 deletion is used as control to confirm the specificity of the PLA signals observed. At least 75 cells were quantified for each condition. Bars indicate the mean values, error bars represent standard errors of the mean, and asterisks indicate statistical significance (*t* test, two-tailed, unpaired). **e, f** EXO1 (**e**)

and MRE11 (**f**) SIRF experiments showing the differential impact of the loss of each of these nucleases on the recruitment to nascent DNA of the other nuclease, upon exposure of HeLa-BRCA2[KO] cells to 0.4 mM HU. Loss of MRE11 suppressed EXO1 binding to nascent DNA, while EXO1 depletion did not affect MRE11 recruitment under these conditions. Depletion of MRE11 or EXO1 respectively is used as control to confirm the specificity of the SIRF signals observed. At least 45 cells were quantified for each condition. Bars indicate the mean values, error bars represent standard errors of the mean, and asterisks indicate statistical significance (*t* test, two-tailed, unpaired). Schematic representations of the assay conditions are shown at the top. Source data are provided as a Source data file.

previously described synthetic lethality between the BRCA pathway and RAD52, an essential component of the BIR machinery[66–68].

Finally, our work suggests that ssDNA gap-induced genomic instability may be relevant for carcinogenesis. We show that exposure to environmental contaminants BPA and DEHP also causes nascent strand ssDNA gaps, which are processed by MRE11 and EXO1 nucleases similar to HU-induced gaps. This is results in DSB formation, which is known to promote chromosomal translocations and other forms of genomic instability. This process may thus contribute significantly to the carcinogenic potential of these agents. An important caveat of our studies is that the treatment conditions employed (200 μM BPA or 200 μM DEHP for 30 min to 3 h), while similar to those used in other cell culture studies[38–40,44,53], are likely to be higher than the

physiological exposure levels of the human population to these agents. Biomonitoring of BPA in human individuals has been typically performed from urine samples, and studies found that the median daily excretion of BPA through urine was 34 ng/kg of body mass[69]. However, it was subsequently found that BPA accumulates in fat tissues and that biomonitoring studies using urine underestimate the total body burden of BPA[70,71]. Daily dietary intake of BPA may be as high as 1.5 μg/kg of body mass[38,72]. Given that the average weight of an American adult is 80 kg, it would take about 377 days to consume 200 μmoles of BPA. Our studies are designed to investigate the acute effects of BPA exposure over control (DMSO) since it is unclear what the exact BPA concentration in various tissues is, and it is impractical to investigate this under physiological exposure conditions (exposure

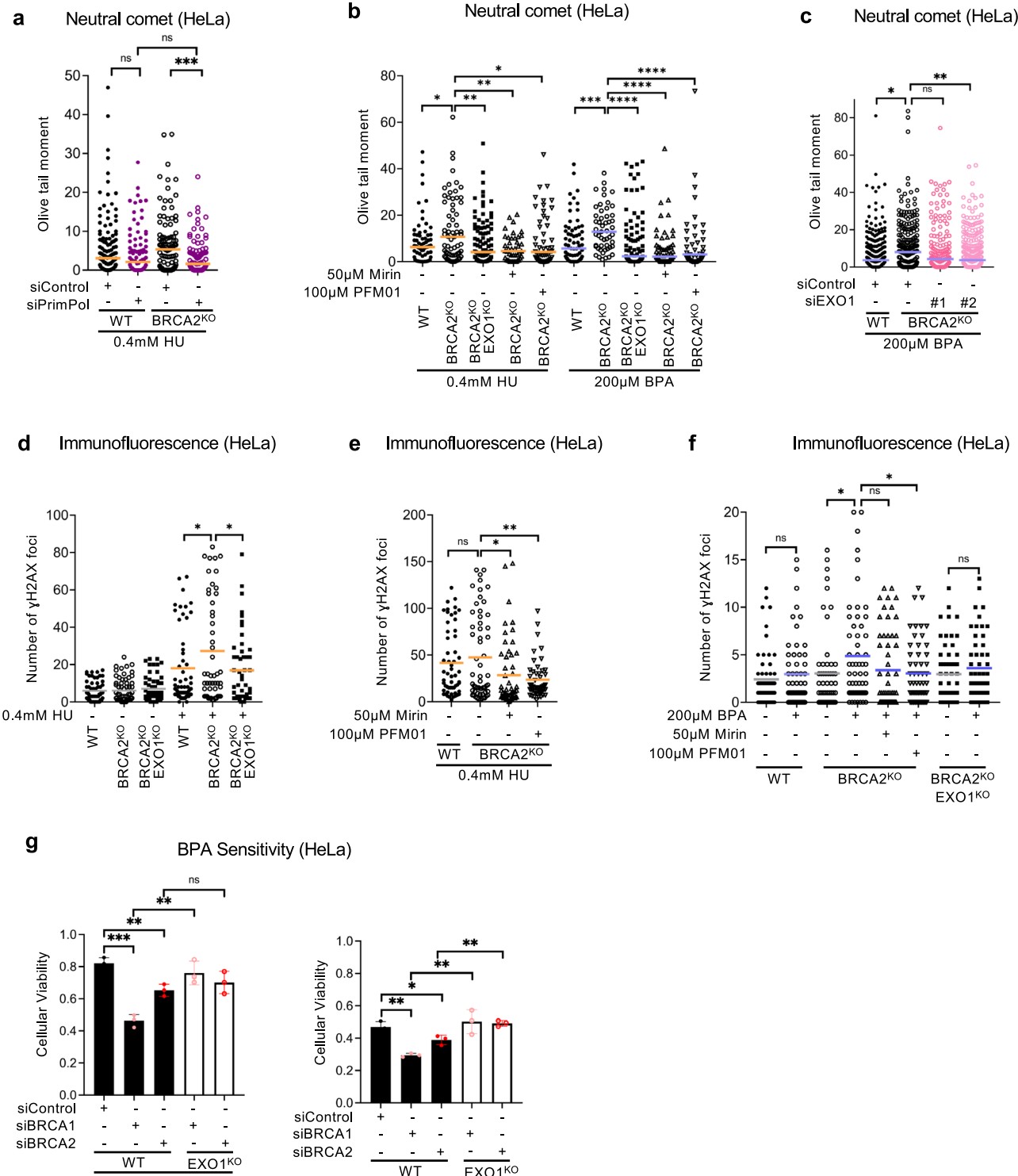

**Fig. 6 | Exonucleolytic and endonucleolytic processing of HU and BPA-induced ssDNA gaps by MRE11 and EXO1 leads to DSB formation. a** Neutral comet assay showing that DSBs induced by treatment of HeLa-BRCA2^KO cells with 0.4 mM HU for 2 h are suppressed by PRIMPOL depletion. At least 70 comets were quantified for each sample. The median values are marked on the graph, and asterisks indicate statistical significance (Mann–Whitney, two-tailed). **b, c** Neutral comet assays showing that treatment with 0.4 mM HU or 200 μM BPA for 2 h causes DSBs in HeLa-BRCA2^KO cells, which are suppressed by inhibition of MRE11 exonuclease activity by mirin or if its endonuclease activity by PFM01 (**b**), EXO1 knockout (**b**) or EXO1 knockdown (**c**). At least 40 comets were quantified for each sample. The median values are marked on the graph, and asterisks indicate statistical significance (Mann–Whitney, two-tailed). **d–f** γH2AX immunofluorescence showing

that that treatment with 0.4 mM HU (**d, e**) or with 200 μM BPA (**f**) for 2 h increases γH2AX foci in in HeLa-BRCA2^KO cells, which is suppressed by EXO1 knockout (**d, f**) or by inhibition of MRE11 exonuclease activity by mirin or if its endonuclease activity by PFM01 (**e, f**). At least 45 cells were quantified for each condition. The mean value is represented on the graphs, and asterisks indicate statistical significance (*t* test two-tailed, unpaired). **g** Cellular viability assays showing that loss of EXO1 partially suppresses the BPA sensitivity of BRCA1- or BRCA2-knockdown HeLa cells, at two different concentrations as indicated. The average of three independent experiments, with standard deviations indicated as error bars, is shown. Asterisks indicate statistical significance (*t* test, two-tailed, unpaired). Source data are provided as a Source data file.

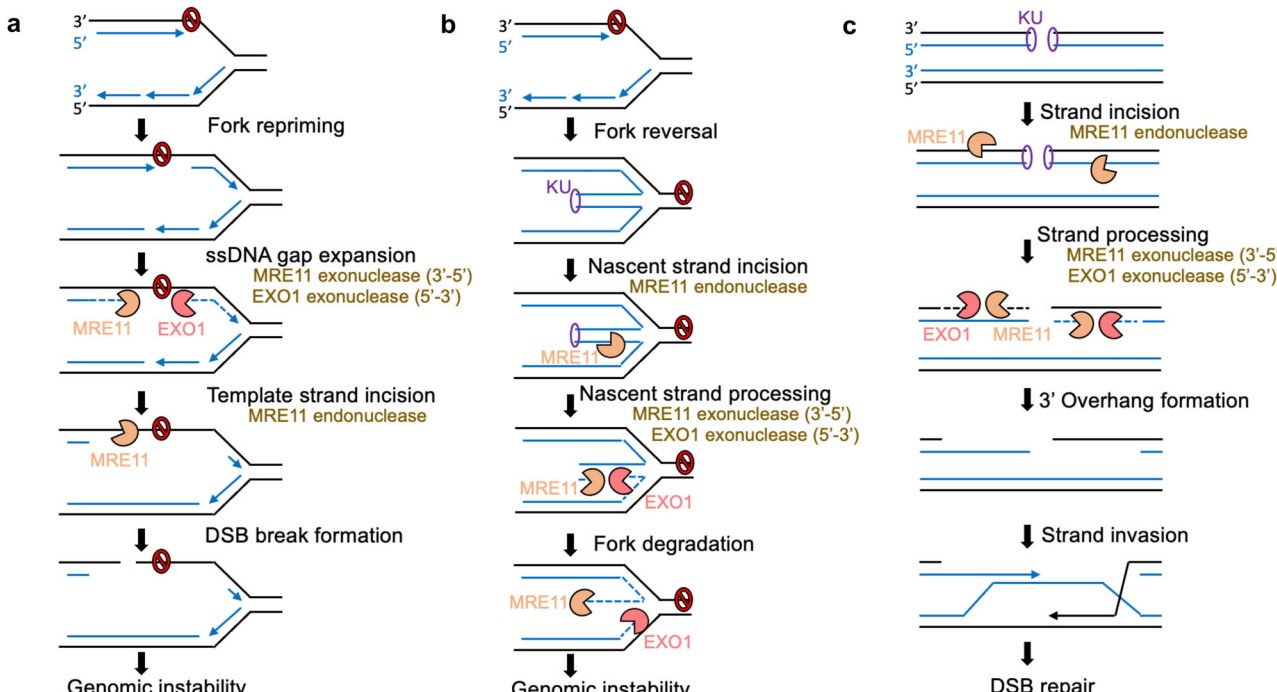

**Fig. 7 | Schematic representation of the proposed models.** The sequential activities of EXO1 and MRE11 exonuclease activities and or MRE11 endonuclease activity in ssDNA gap formation (**a**), degradation of reversed forks (**b**), and end resection during DSB repair by HR (**c**) are shown. In all cases, the MRE11 3′–5′ exonuclease activity and the EXO1 5′–3′ exonuclease activity bidirectionally expand a ssDNA nick or short gap. In ssDNA gap processing (**a**), the MRE11 endonuclease cleaves the template strand to cause a DSB subsequent to gap expansion. In fork degradation (**b**) and DSB end resection (**c**), the DSB ends are protected by binding of the KU complex, which prevents the direct engagement of exonuclease activities. The endonuclease activity of MRE11, which is recruited to dsDNA in the proximity of the KU-bound DSB end, is required to create the initial nick extended by the exonuclease activities or EXO1 and MRE11. MRE11-mediated exonucleolytic degradation towards the DSB end results in removal of the KU complex, while EXO1 catalyzes long-range resection in the opposite direction (moving away from the DSB end).

to lower doses over several years) using the specific assays for measuring ssDNA gap processing. Nevertheless, our findings are important to raise awareness to the possible outcomes of BPA exposure, considering how widely spread its usage is, and how human exposure to BPA may only increase in the future.

Unlike HU treatment, BPA and DEHP exposures caused DNA gaps in normal (BRCA-proficient cells) and not only in BRCA-deficient cells, in line with their broad carcinogenic potential[34–37]. Nevertheless, ssDNA gap induction by BPA and DEHP appeared to be more pronounced in BRCA-deficient cells. While exposure to these agents has been associated with breast and ovarian cancers among others, to our knowledge a hypersensitivity of BRCA mutation carriers to BPA or DEHP-induced carcinogenesis has not been reported. Nevertheless, our work suggests that BRCA mutation carriers may potentially benefit from limiting exposure to these agents.

## Methods
### Cell culture and protein techniques
HeLa (obtained from ATCC) and RPE1 cells were grown in Dulbecco's modified Eagle's media (DMEM). DLD-1 cells were obtained from Dr. Robert Brosh (National Institute on Aging, Baltimore, MD) and were grown in Roswell Park memorial Institute (RPMI) 1640 media. Media was supplemented with 15% FBS and penicillin/streptomycin. HeLa-BRCA2$^{KO}$ cells[73] and HeLa-BRCA2$^{KO}$EXO1$^{KO}$ cells[27] were generated in our laboratory and previously described. RPE1 and RPE1-BRCA1$^{KO}$ cells (also harboring p53 homozygous deletion) were obtained from Dr. Alan D'Andrea (Dana-Farber Cancer Institute, Boston, MA)[50]. To knockout EXO1 in HeLa cells, a commercially available CRISPR/Cas9 KO plasmid (Santa Cruz Biotechnology sc-402356) was used.

Transfected cells were FACS-sorted into 96-well plates using a BD FACSAria II instrument. Resulting colonies were screened by Western blot.

Gene knockdown was performed using Lipofectamine RNAiMAX. AllStars Negative Control siRNA (Qiagen 1027281) was used as control. The following oligonucleotide sequences (Stealth or SilencerSelect siRNA, ThermoFisher) were used:

BRCA1: AAUGAGUCCAGUUUCGUUGCCUCUG; BRCA2: AUUAG-GAGAAGACAUCAGAAGCUUG; MRE11: AAUAACUCGAGGCAGGUAU-GUAAUG; EXO1#1: CCUGUUGAGUCAGUAUUCUCUUUCA (used unless otherwise specified); EXO1#2: Assay ID s52594; ZRANB3: UGGCAAU-GUAGUCUCUGCACCUAUA; SMARCAL1: CACCCUUUGCUAACC-CAACUCAUAA; PRIMPOL: Assay ID s47418.

Denatured whole cell extracts were prepared by boiling cells in 100 mM Tris, 4% SDS, 0.5 M β-mercaptoethanol. Antibodies used for Western blot, at 1:500 dilution, were:

BRCA1 (Santa Cruz Biotechnology sc-6954); BRCA2 (Bethyl A303-434A); MRE11(GeneTexGTX70212); EXO1 (Novus NBP2-16391);ZRANB3 (Invitrogen PA5-65143); SMARCAL1 (Invitrogen PA5-54181); PRIMPOL (Proteintech 29824-1-AP); Cleaved Caspase-3 (Cell Signaling Technology 9664); GAPDH (Santa Cruz Biotechnology sc-47724); Vinculin (Santa Cruz Biotechnology sc-73614).

Chemical compounds used were: BPA (MilliporeSigma 239658), DEHP (MilliporeSigma 67261), mirin (Selleck Chemicals S8096), PFM01 (Tocris 6222), Z-VAD-FMK (Santa Cruz Biotechnology sc-3067).

### Functional assays
Neutral and BrdU alkaline comet assays were performed[22] using the Comet Assay Kit (Trevigen, 4250-050). For the BrdU alkaline comet assay, cells were incubated with 100 µM BrdU as indicated. Chemical

compounds (0.4 mM HU, 50 μM mirin, 100 μM PFM01, 200 μM BPA, 200μM DEHP) were added according to the labeling schemes presented. Slides were stained with anti-BrdU (BD 347580) antibodies and secondary AF568-conjugated antibodies (Invitrogen A-11031). Slides were imaged on a Nikon microscope operating the NIS Elements V1.10.00 software. Olive tail moment was analyzed using CometScore 2.0. Immunofluorescence was performed[74] using a γH2AX antibody (MilliporeSigma JBW301). Slides were imaged on a confocal microscope (Leica SP5) and analyzed using ImageJ 1.53a software.

### Drug sensitivity assays

To assess cellular viability upon drug treatment, a luminescent ATP-based assay was performed using the CellTiterGlo reagent (Promega G7572) according to the manufacturer's instructions. Following treatment with siRNA, 1500 cells were seeded per well in 96-well plates and incubated with the indicated doses of HU, BPA or cisplatin for 3 days. Luminescence was quantified using a Promega GloMax Navigator plate reader. Apoptosis assays were performed using the FITC Annexin V kit (Biolegend, 640906). Quantification was performed on a BD FACS-Canto 10 flow cytometer using the FlowJo v10 software.

### DNA fiber combing assays

Cells were incubated with 100 μM IdU and 100 μM CldU as indicated. Chemical compounds (4 mM or 0.4 mM HU, 50 μM mirin, 100 μM PFM01, 200 μM BPA, 200 μM DEHP) were added according to the labeling schemes presented. Next, cells were collected and processed using the FiberPrep kit (Genomic Vision EXT-001) according to the manufacturer's instructions. Samples were added to combing reservoirs containing MES solution (2-(N-morpholino) ethanesulfonic acid) and DNA molecules were stretched onto coverslips (Genomic Vision COV-002-RUO) using the FiberComb Molecular Combing instrument (Genomic Vision MCS-001). For S1 nuclease assays, MES solution was supplemented with 1 mM zinc acetate and either 40 U/mL S1 nuclease (ThermoFisher 18001016) or S1 nuclease dilution buffer as control, and incubated for 30 minutes at room temperature. Slides were then stained with antibodies detecting CldU (Abcam 6236) and IdU (BD 347580), and incubated with secondary Cy3 (Abcam 6946) or Cy5 (Abcam 6565) conjugated antibodies. Finally, the cells were mounted onto coverslips and imaged using a confocal microscope (Leica SP5) and analyzed using LASX 3.5.7.23225 software.

### Proximity ligation-based assays

For PLA assays, cells were seeded into 8-chamber slides and 24 h later, were treated with 0.4 mM HU or 200 μM BPA for 3 h as indicated. Cells were then permeabilized with 0.5% Triton for 10 min at 4 °C, washed with PBS, fixed at room temperature with 3% paraformaldehyde in PBS for 10 min, washed again in PBS and then blocked in Duolink blocking solution (Millipore Sigma DUO82007) for 1 h at 37 °C, and incubated overnight at 4 °C with primary antibodies. Antibodies used were: MRE11 (Genetex GTX70212) and EXO1 (Novus NBP2-16391). Samples were then subjected to a proximity ligation reaction using the Duolink kit (Millipore Sigma DUO92008) according to the manufacturer's instructions. Slides were imaged using a Deltavision microscope with SoftWorx 6.5.2 software, and images were analyzed using ImageJ 1.53a software.

For SIRF assays, cells were seeded into 8-chamber slides and 24 h later they were pulse-labeled with 50 μM EdU and treated with chemical compounds (4 mM or 0.4 mM HU, 200 μM BPA, 200μM DEHP) according to the labeling schemes presented. Cells were permeabilized with 0.5% Triton for 10 min at 4 °C, washed with PBS, fixed at room temperature with 3% paraformaldehyde in PBS for 10 min, washed again in PBS, and then blocked in 3% BSA in PBS for 30 min. Cells were then subjected to Click-iT reaction with biotin-azide using the Click-iT Cell Reaction Buffer Kit (ThermoFisher C10269) for 30 min

and incubated overnight at 4 °C with primary antibodies diluted in PBS with 1% BSA. The primary antibodies used were: Biotin (mouse: Jackson ImmunoResearch 200-002-211; rabbit: Bethyl Laboratories A150-109A); MRE11 (GeneTex GTX70212); EXO1 (Santa Cruz Biotechnology sc-56092). Next, samples were subjected to a proximity ligation reaction using the Duolink kit (MilliporeSigma DUO92008) according to the manufacturer's instructions. Slides were imaged using a Deltavision microscope with SoftWorx 6.5.2 software, and images were analyzed using ImageJ 1.53a software. To account for variation in EdU uptake between samples, for each sample, the number of protein-biotin foci were normalized to the average number of biotin-biotin foci for that respective sample. The scale bars for the SIRF micrographs shown represent 10 μm.

### Statistics and reproducibility

For immunofluorescence, Annexin V, SIRF and PLA assays, as well as CellTiterGlo cellular viability assays the $t$ test (two-tailed, unpaired) was used. For DNA fiber assays and comet assays the Mann–Whitney statistical test (two-tailed) was performed. For immunofluorescence, DNA fiber combing, PLA, SIRF, and comet assays, results from one experiment are shown; the results were reproduced in at least one additional independent biological conceptual replicate. Western blot experiments were reproduced at least two times. Statistical analyses were performed using GraphPad Prism 10 and Microsoft Excel v2205 software. Statistical significance is indicated for each graph (ns = not significant, for $p > 0.05$; * for $p \leq 0.05$; ** for $p \leq 0.01$; *** for $p \leq 0.001$, **** for $p \leq 0.0001$).

### Reporting summary

Further information on research design is available in the Nature Portfolio Reporting Summary linked to this article.

## Data availability

The data supporting the findings of this study are available within the paper and its Supplementary Information. Source data are provided with this paper.

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

## Acknowledgements

We would like to thank Dr. Robert Brosh and Dr. Alan D'Andrea for materials and the Penn State College of Medicine Imaging (RRID:SCR-021200) and Flow Cytometry (RRID:SCR-021134) core facilities. This work was supported by NIH R01ES026184 and NIH R01GM134681 (to G.-L.M.) and NIH R01CA244417 (to C.M.N.).

## Author contributions

A.H., A.D., J.S., C.M.N. and G.-L.M. designed and conducted the experiments and wrote the paper.

## Competing interests

The authors declare no competing interests.
