## [Peer Review File · Nature Communications]

Multi-step processing of replication stress-derived nascent strand DNA gaps by MRE11 and EXO1 nucleasesREVIEWER COMMENTS

Reviewer #1 (Remarks to the Author):

In their manuscript, Hale and co-workers study the role of MRE11 and EXO1 in ssDNA gap processing.

The appearance of ssDNA gaps has recently received increasing amounts of attention, through -among others- the work of the Vindigni lab, who demonstrated ssDNA gaps in BRCA-deficient cells, and the work of the Cantor lab who linked ssGAP formation to the sensitivity to DNA damaging agents as part of the BRCAness phenotype.

In similarity of processing of stalled forks in BRCA-deficient cells by both MRE11 and EXO1, the authors investigate whether MRE11 and EXO1 are also involved in processing of ssDNA gaps. On a somewhat separate topic, the authors investigate BPA and DEHP, two environmental toxic agents that have been linked to carcinogenesis.

Overall, the authors show convincingly that forks are processed by both MRE11 and EXO1, and that BPA and DEHP also cause ssDNA gaps, which are processed similarly when compared to HU-induced gaps. Although the results are clear, the impact of these mechanisms of genotoxicity or mutagenicity remain unclear, and with that, the knowledge increase is somewhat incremental.

Comments:

1. Line 54/line 94-96: In the introduction it is mentioned that fork degradation/hyper-resection results in sensitivity to genotoxic agents, and later it is mentioned that ssDNA gap accumulation is even better correlated to chemosensitivity. It would be good to also cite and discuss the work of the Jasin lab, who showed that it is the HR function, not the fork protection role of BRCA2 that mediates sensitivity to PARP inhibition (Feng and Jasin, Nat Comm, 2017). In more recent work that has been presented at meetings, the Jasin lab has presented similar work using separation-of-function mutants, in which ssGAP formation was shown to not be responsible for PARP inhibitor sensitivity.

2. It is unclear to me how if the analysis of fiber tracts reflects multiple independent experiments of whether a single experiment has been performed.

3. If MRE11 and EXO1 process ssDNA gaps in opposite direction, why does inactivation of either protein result in a near-complete loss of ssDNA formation? I would expect a partial effect upon depletion of either protein, reflecting the 3' or 5' directed processing.

4. To what extent is processing of ssDNA gaps by MRE11 and EXO1 relevant for cellular or genetic outcomes. All assays show processing of ssDNA gaps, but it remains unclear what the physiological impact of these mechanisms is.

5. BPA and DEHP induce ssDNA gaps, which are processed similarly when compared to low dose HU-induced gaps. However, are the ssDNA gaps induced by BPA and DEHP also the DNA lesions that are connected to mutagenesis or cytotoxicity?

Reviewer #2 (Remarks to the Author):

In this manuscript by Hale et al., the authors presented data that MRE11 and EXO1 are required for

stress-induced ssDNA gap formation in newly replicated DNA which are suppressed by the BRCA pathway. They also showed that environmental toxicants such as bisphenol A (BPA) and diethylhexyl phthalate (DEHP) can also induce ssDNA gaps on nascent DNA which require MRE11 and EXO1. The topic is interesting and relevant to cancer initiation and chemotherapy responses. The data are largely convincing. However, they lack sufficient novelty, rigor, and depth to support the model proposed by the authors (Fig.6), and substantially advance our current understanding of the mechanism and therapeutic relevance of ssDNA gaps.

Major points:

1. The notion that MRE11 and EXO1 cooperate to bidirectionally expand ssDNA gaps is not new. It is well-known in the context of end-resection during DSB repair, and more recently suggested by the authors themselves to promote nascent DNA degradation in BRCA-deficient/KU-proficient cells upon stress-induced fork reversal. Here the authors proposed that a similar cooperation between MRE11 and EXO1 also occurs on stressed (but unreversed) nascent DNA to bidirectionally enlarge the ssDNA nicks (generated by repriming) followed by the cutting of template DNA strand by MRE11 to generate DSB. From a conceptual standpoint, the novelty of this study is not very high (i.e. same cooperative relationship between MRE11 and EXO1 in different DNA contexts) though the model proposed in the end contains novel elements regarding the sequential involvement of the exonuclease and endonuclease activities of MRE11 (which is opposite from DSB end resection) and BIR repair of the gap-derived DSB. Nonetheless, both of these novel aspects are speculative and no data in the paper can be found to directly support them.
2. Piberger et al. (Nature Commun. 2020) reported similar findings that BPDE-induced nascent DNA gaps require MRE11 and EXO1 activities and proposed a similar model that these two nucleases bidirectionally expand PrimPol-generated ssDNA gaps followed by RAD51 binding. Though the DNA replication stressor is different (BPDE vs. HU, BPA, and DEHP), the idea that these two nucleases work together to enlarge the gaps on nascent DNA of the leading strand behind stressed forks is similar.
3. It is unclear how the authors, based on the assays used in the paper, can conclude that the ssDNA gaps observed by the BrdU comet assay and S1 DNA fiber analysis are not on the reversed nascent DNA. It seems that the authors assumed that 0.4mM HU does not induce fork reversal based on the lack of detectable fork degradation phenotype (CldU/IdU ratio) in BRCA-deficient cells. However, Zellweger et al. (JCB, 2015) showed that 1hr treatment of U2OS and RPE-1 cells with 0.5mM HU triggered high frequency (>20%) of fork reversal.
4. The sample sizes for all the DNA fiber experiments are too small. Even though the combing assay was used, ~50 fibers per condition are not rigorous enough, especially when assessing subtle changes or ruling out effects.

Minor points:

1. Same cell lines should be used to determine the loss of function effects of MRE11 and EXO1 in the contexts of BRCA1 and BRCA2 deficiencies. For instance, in Fig.1, HeLa cells were used only for BRCA2 loss and RPE1 cells were only used for BRCA2 loss.
2. Many experiments lack untreated controls to confirm the phenotypes are stress-dependent as the authors claimed. For instance, for all the S1 DNA fiber experiments (Fig.1D-E, 3J-L, 4C-D) untreated control cells should also be included and analyzed to determine whether the CldU/IdU ratio difference depends on the drugs. It is possible that intrinsic difference in gap formation during unperturbed replication exists between control cells and those lacking BRCA1/2 and/or MRE11 and EXO1.
3. In Fig.2B-E, it would be more rigorous to test MRE11 and EXO1 recruitment to nascent DNA using both schemes instead of one scheme for one nuclease only. If the assay can indeed detect the recruitment of the two nucleases to opposite ends of the gaps (5' vs. 3'), this will serve to further test it.

4. In Fig.2C and 2F, why did the authors perform statistical analysis in WT cells for EXO1 SIRF but not MRE11 SIRF? In fact, 0.4mM HU in Fig.2C causes visible decrease of MRE11/EdU interaction though no stats are performed. Is this real and if so why?

5. Based on the DNA fiber data in Fig.1, it seems 0.4mM HU for 1hr is quite permissive to replication (though slower than untreated condition as inferred from the CldU/IdU ratio). Thus, 0.4mM HU treatment for 3hr (used in many subsequent experiments e.g. Fig.2B) is expected to generate more nascent DNA. If so, how can the authors be sure that the initially EdU-labeled DNA is still "nascent" after the 3hr 0.4mM HU treatment?

6. In the abstract and end of introduction, the authors wrote that "the nascent strand" at the ssDNA gap is cleaved by the MRE11 endonuclease generating DSBs. Do the authors mean "template" or "parental" strand?

7. The MRE11 inhibitor concentrations (e.g. 50uM mirin) seem unnecessarily high.

8. No background information was given as to the chosen concentrations of BPA and DEHP.

Reviewer #3 (Remarks to the Author):

This study by Hale et al. investigates the mechanisms through which ssDNA gaps left behind DNA replication forks (RF) are extended by the MRE11 and EXO1 nucleases. The authors also present data regarding the potential of bisphenol A (BPA) and diethylhexyl phthalate (DEHP), which are used in plastics manufacturing, to generate ssDNA gaps.

Data indicating that BRCA1/2 prevent the extension of ssDNA gaps by MRE11 and EXO1 is first presented. These results are conceptually consistent with previously published results (e.g., Piberger et al. *Nature Communications* volume 11, Article number: 5863 (2020)), the difference here being that ssDNA gaps are induced by HU in BRCA1/2-null cells vs bulky adducts. While the data is of good quality, the results were somewhat expected in my opinion. I note that several researchers now present only the second label in S1 nuclease assays, rather than the CldU/IdU ratio which can be influenced by RF progression speed prior to exposure to genotoxins. I think it would be beneficial, throughout the manuscript, to present the length of each analog-labeled tracks in addition to the ratio.

The authors then aim to demonstrate that MRE11 and EXO1 are recruited to ssDNA gaps. In figure 2A, they claim that using 0.4 mM vs 4 mM HU does not cause RF reversal and instead mostly generates ssDNA gaps, based on the absence of "RF protection defect". In my opinion, these experiments are insufficient to ensure that little/no RF reversal exist in the presence of 0.4 mM HU. First, previous evidence (Zellweger et al. *J Cell Biol.* 2015 Mar 2; 208(5): 563–579) clearly indicate that 0.5 mM HU causes RF reversal, at least in U2OS cells. Moreover, if I understand the labeling scheme correctly, CldU is not present during the incubation with HU. It is expected that RF will progress in 0.4 mM HU; indeed, data presented in Fig. 1D-E indicate that 0.4mM HU does not completely prevent RF progression. Therefore, RF will progress and may then reverse at some point beyond the CldU labeled track: any RF protection defect in these conditions should be undetectable since it will occur in non-labeled DNA. Even if DNA was still labeled, RF progression would render the detection of eventual RF protection defects difficult since progression and protection defects would counteract each other.

Because of the above, in my opinion the SIRF experiments used to claim that MRE11 and EXO1 are recruited to ssDNA gaps, and not to reversed RF, are inconclusive. Indeed, lesser amount of RF reversal followed by nucleolytic activity at the DSB end of the reversed fork might be sufficient to generate, or at least contribute to, the SIRF signals in Figure 2B-G. In my opinion, it is necessary to abolish RF reversal, e.g., using knock-down e.g. of SMARCAL1, and to demonstrate that SIRF signals

are still present. Moreover, depletion of PRIMPOL should also reduce the signals in these experiments if indeed they depend on the generation of ssDNA gaps.

The authors then show that BPA and DEHP causes DNA damage (Figure 3A-I); I note a certain degree of disconnect between these results and the first figures, which seem like a different story. While of good quality, these data are not particularly novel as they are in line with several previously published reports (e.g., PMID: 27923681, PMID: 27208089, PMID: 28771232). More novel is the notion that treatment with these agents causes ssDNA gap formation. However, I question the relevance of the doses used; the authors should justify whether these doses are physiologically relevant as they seem quite high. Are these doses close to what happens in organisms affected by BPA/DEHP residues? Importantly, throughout the experiments of Figure 3, the authors mention that BPA and DEHP are particularly deleterious to BRCA-deficient cells; however, no statistical analyses are presented to compare the extent of the effect of the compounds in BRCA1/2-null vs WT cells.

The authors then present data indicating that MRE11 and EXO1 influence ssDNA gap extension, which is again in line with previously published data using other genotoxins (Piberger et al. Nature Communications volume 11, Article number: 5863 (2020)). In Page 12, the authors state that depletion of MRE11 or of EXO1 suppressed nascent ssDNA accumulation, however, this is not directly shown for EXO1 in my opinion; a BrdU alkaline comet assay (as Figure 4B) or similar experiments would be appropriate.

The authors go on to use SIRF to state that MRE11 and EXO1 are recruited to ssDNA gaps caused by BPA. However, I think that a similar argument to the one made with regards to 0.4 vs 4 mM HU holds in this case as well. 200 uM BPA clearly permits RF progression (Fig. 4C), and therefore the experiment in Fig. 3G is insufficient in my opinion to state that no RF reversal or degradation occurs in presence of this compound. If RF progress during the incubation with the drugs, then any degradation ("RF protection defect") might be compensated by elongation, rendering the experiment inconclusive. Depletion of PRIMPOL and of RF reversal factors (SMARCAL, ZRANB3, etc...) would provide stronger evidence to link the recruitment of MRE11 and EXO1 specifically at ssDNA gaps and not at reversed forks.

Finally, the authors study the possibility that extended gaps might be processed into DSBs by MRE11 endonuclease activity. To this end, they use mirin, but this inhibitor prevents ssDNA gaps extension. While PFM01 inhibits the endonuclease activity of MRE11, it would be important to demonstrate that PFM01 treatment does not influence gap extension in their experimental conditions. Complementation experiments with MRE11 mutants specifically defective in exo vs endonuclease activity (e.g., PMID: 32246962) would also be appropriate. I'm also not convinced that the evidence presented is sufficient to claim that MRE11 endonuclease activity acts specifically at ssDNA gaps; it is formally possible that it acts elsewhere, e.g., at stalled RF. More evidence showing that MRE11-induced DSBs occur specifically at ssDNA gaps behind RF are necessary. It would also be worthwhile to exclude apoptosis-associated DSBs in their experiments, as they cite articles mentioning this possibility in the introduction. Finally, in their discussion they propose that HR might not be functional at ssDNA gaps-associated DSBs; while this is interesting speculation, they show no data supporting this.

Minor point:

Page 3: "Arrested forks can also be restarted by initiating de novo DNA synthesis downstream of the lesion, upon repriming by factors such as the primase-polymerase PRIMPOL and, on the leading strand, the Pol α -primase complex".

I assume that the authors mean lagging strand here?

Page 7: "These findings suggest a general role for EXO1 in ssDNA gap expansion.

And for MRE11 presumably since the previous paragraph states essentially the same for this nuclease.

Response to referees

We would like to thank the reviewers for their helpful and constructive comments, which led to a significantly improved manuscript. To address the reviewers' concern, we are submitting a substantially revised manuscript with 35 new figure panels, as well as 2 revised figure panels.

In the revised manuscript, we present additional results that further extend the relevance of our work. In particular, we address the following two major points raised by the reviewers:

- *Functional relevance of ssDNA gap processing on cellular cytotoxicity:*
 - We expanded our studies to cisplatin, by showing that the ssDNA gaps induced by treatment with this chemotherapeutic are also expanded by EXO1 and MRE11.
 - We show that the processing of ssDNA gaps induced by HU and BPA are associated with cellular cytotoxicity. Specifically, we report that loss of EXO1 can partially suppress the HU, cisplatin and BPA sensitivity of BRCA1- or BRCA2-depleted cells, suggesting that ssDNA gap expansion may be partially associated with cellular sensitivity to DNA damaging drugs.
- *Mechanistic insights into gap processing by EXO1 and MRE11*
 - We show that MRE11 and EXO1 engage ssDNA gaps induced by HU and BPA in a co-regulated manner. This extended analysis is presented in two whole new figures (Figure 5 and Supplementary Figure S7), and also serves to better integrate the two parts of our manuscript since the findings are similar with HU and BPA.
 - We show that the processing of ssDNA gaps by EXO1 and MRE11 is not affected by the of fork reversal, implying that it occurs behind replication forks, and not on reversed forks.

We would also like to point out a change in the title of the manuscript. The way the term “*sequential*” was used in the title of our original submission may give the reader the impression that MRE11 and EXO1 act one after another. This is true in the sense that the MRE11 endonuclease activity acts after exonuclease-mediated gap expansion. However, our original and new data indicate that the exonucleases activities of MRE11 and EXO1 act in a coordinated manner and likely at the same time. Thus, the term “*sequential*” may be misleading. We thus replaced it with the term “*multi-step*” in the revised article title, which we think better captures the process we describe in the manuscript.

Reviewer #1

In their manuscript, Hale and co-workers study the role of MRE11 and EXO1 in ssDNA gap processing.

The appearance of ssDNA gaps has recently received increasing amounts of attention, through -among others- the work of the Vindigni lab, who demonstrated ssDNA gaps in BRCA-deficient cells, and the work of the Cantor lab who linked ssGAP formation to the sensitivity to DNA damaging agents as part of the BRCAness phenotype.

In similarity of processing of stalled forks in BRCA-deficient cells by both MRE11 and EXO1, the

authors investigate whether MRE11 and EXO1 are also involved in processing of ssDNA gaps. On a somewhat separate topic, the authors investigate BPA and DEHP, two environmental toxic agents that have been linked to carcinogenesis.

Overall, the authors show convincingly that forks are processed by both MRE11 and EXO1, and that BPA and DEHP also cause ssDNA gaps, which are processed similarly when compared to HU-induced gaps. Although the results are clear, the impact of these mechanisms of genotoxicity or mutagenicity remain unclear, and with that, the knowledge increase is somewhat incremental.

We thank the reviewer for their comments. Please find below our answers to these comments:

Comments:

1. Line 54/line 94-96: In the introduction it is mentioned that fork degradation/hyper-resection results in sensitivity to genotoxic agents, and later it is mentioned that ssDNA gap accumulation is even better correlated to chemosensitivity. It would be good to also cite and discuss the work of the Jasin lab, who showed that it is the HR function, not the fork protection role of BRCA2 that mediates sensitivity to PARP inhibition (Feng and Jasin, Nat Comm, 2017). In more recent work that has been presented at meetings, the Jasin lab has presented similar work using separation-of-function mutants, in which ssGAP formation was shown to not be responsible for PARP inhibitor sensitivity.

We thank the reviewer for pointing this out, and we apologize for omitting to discuss this work in the original submission. We agree with the reviewer that the connection between ssDNA gap accumulation and chemotherapy sensitivity is not clear, and it is the focus of intense research in the field, with a number of alternative models being put forward. In the revised manuscript (page 5), we discuss the Feng and Jasin manuscript indicated by the reviewer, as well as the new work from the Jasin lab recently posted on bioRxiv, which indicate that BRCA2 promotes chemosensitivity primarily through HR. We think that our work describing the processing of ssDNA gaps into DSBs is relevant in this context as well, considering the role of HR in DSB repair.

2. It is unclear to me how if the analysis of fiber tracts reflects multiple independent experiments of whether a single experiment has been performed.

As we described in the "Statistics and reproducibility" paragraph of the Methods section, in general results from one experiment are shown; the results were reproduced in at least one additional independent biological replicate.

3. If MRE11 and EXO1 process ssDNA gaps in opposite direction, why does inactivation of either protein result in a near-complete loss of ssDNA formation? I would expect a partial effect upon depletion of either protein, reflecting the 3' or 5' directed processing.

We agree with the reviewer that this is somewhat puzzling. To address this, we thought to perform epistasis analyses with MRE11 and EXO1. In the S1 nuclease DNA fiber combing assay, which measures the presence of ssDNA gaps in individual replication tracts, loss of either MRE11 or EXO1 completely suppressed the reduction in CldU/IdU ratios caused by S1 treatment, indicating that ssDNA gaps measured by this assay are equally restrained by inactivation of either nuclease -as pointed out by the reviewer. This complete suppression did

not allow us to perform epistasis analyses with MRE11 and EXO1. In the revised manuscript, we show such epistasis analyses using the BrdU alkaline comet assay, whose readout reflects the overall amount of replication-derived ssDNA in cells. We found that, with both HU and BPA, treatment with mirin did not further reduce the comet tail moment in EXO1-depleted cells, indicating that EXO1 and MRE11 are epistatic for ssDNA gap suppression (**new Fig. 5a,b**). To further investigate this, in proximity ligation assays (PLA) we found that MRE11 and EXO1 interact upon induction of ssDNA gaps with either HU or BPA (**new Fig. 5c,d**), suggesting that they may engage ssDNA as a complex. In line with this, using SIRF assays we now show that MRE11 depletion suppresses EXO1 recruitment to ssDNA upon 0.4mM HU treatment (**new Fig. 5e**). Similarly, depletion of PRIMPOL, which is responsible for ssDNA gap formation, also suppresses EXO1 recruitment (**new Fig. 2h**). Overall, these findings indicate that EXO1 engagement on ssDNA gaps depends on MRE11.

In contrast, we found that MRE11 recruitment to nascent DNA upon 0.4mM HU treatment does not depend on EXO1 (**new Fig. 5f**) suggesting that MRE11 can be recruited to ssDNA gaps independently of EXO1, but cannot exert its gap expansion activity in the absence of EXO1. However, this result may also reflect MRE11 binding to nascent DNA at other structures than ssDNA gap, since its recruitment is also not suppressed by PRIMPOL depletion (**new Supplementary Fig. 7c**). Overall, our new studies presented in the revised manuscript suggest that MRE11 and EXO1 engage ssDNA gaps in a co-regulated manner, potentially explaining why inactivation of each of them results in a near-complete loss of ssDNA formation.

4. To what extent is processing of ssDNA gaps by MRE11 and EXO1 relevant for cellular or genetic outcomes. All assays show processing of ssDNA gaps, but it remains unclear what the physiological impact of these mechanisms is.

We would like to respectfully point out that our assays measure not only ssDNA gaps, but also DSB accumulation. Nevertheless, we agree with the reviewer that our original manuscript did not address the impact of ssDNA gap processing on cellular sensitivity to drugs inducing ssDNA gaps. The impact of MRE11 is difficult to assess due to pleiotropic effects caused by the multiple roles of MRE11 in genome stability (such as its function in DNA damage detection as part of the MRN complex). Therefore, we focused the sensitivity analyses on EXO1. In the revised manuscript, we show that loss of EXO1 can partially suppress the HU sensitivity of BRCA1- or BRCA2-depleted cells (**new Fig. 1h**), suggesting that ssDNA gap processing may partly account for this sensitivity. Moreover, we extended our analyses to cisplatin, another drug which was shown to induce ssDNA gaps. We found that cisplatin-induced gaps are also expanded by MRE11 and EXO1 (**new Fig. 1d**). Moreover, loss of EXO1 suppressed the cisplatin sensitivity of BRCA1- or BRCA2-depleted cells (**new Fig. 1i**). Overall, these findings suggest that ssDNA gap expansion may be partially associated with cellular sensitivity to DNA damaging drugs, at least under the circumstances investigated here.

5. BPA and DEHP induce ssDNA gaps, which are processed similarly when compared to low dose HU-induced gaps. However, are the ssDNA gaps induced by BPA and DEHP also the DNA lesions that are connected to mutagenesis or cytotoxicity?

Measuring BPA or DEHP-induced mutagenesis in a rigorous manner is notoriously difficult. While previous studies have described the detection of BPA adducts on DNA, to our knowledge BPA-induced mutagenesis was only indirectly inferred from mutational signatures (references 39-47 in our manuscript). Moreover, these studies investigated BPA-induced mutagenicity from

the perspective of mutagenic lesion bypass. Our work suggests that, in addition to this type of point mutagenesis, BPA-induced adducts may also cause genome instability because of gap formation and expansion. As we show in our manuscript, this ultimately causes DSBs, which would potentially lead to NHEJ-mediated deletions and translocations. Measuring this specific mutagenic outcome is difficult and the few available technologies are complicated, time consuming and unreliable.

In the revised manuscript, we are however able to provide insights into how the processing of BPA-induced ssDNA gaps by exonucleases affects cytotoxicity. We show that BRCA1 or BRCA2 depletion causes increased cellular sensitivity to BPA; This increase is suppressed by deletion of EXO1 (**new Fig. 6g**). These findings argue that expansion of BPA-induced ssDNA gaps may contribute to the cellular sensitivity to this agent.

Reviewer #2

In this manuscript by Hale et al., the authors presented data that MRE11 and EXO1 are required for stress-induced ssDNA gap formation in newly replicated DNA which are suppressed by the BRCA pathway. They also showed that environmental toxicants such as bisphenol A (BPA) and diethylhexyl phthalate (DEHP) can also induce ssDNA gaps on nascent DNA which require MRE11 and EXO1. The topic is interesting and relevant to cancer initiation and chemotherapy responses. The data are largely convincing. However, they lack sufficient novelty, rigor, and depth to support the model proposed by the authors (Fig.6), and substantially advance our current understanding of the mechanism and therapeutic relevance of ssDNA gaps.

We thank the reviewer for their comments. Please find below our answers to these comments:

Major points:

1. The notion that MRE11 and EXO1 cooperate to bidirectionally expand ssDNA gaps is not new. It is well-known in the context of end-resection during DSB repair, and more recently suggested by the authors themselves to promote nascent DNA degradation in BRCA-deficient/KU-proficient cells upon stress-induced fork reversal. Here the authors proposed that a similar cooperation between MRE11 and EXO1 also occurs on stressed (but unreversed) nascent DNA to bidirectionally enlarge the ssDNA nicks (generated by repriming) followed by the cutting of template DNA strand by MRE11 to generate DSB. From a conceptual standpoint, the novelty of this study is not very high (i.e. same cooperative relationship between MRE11 and EXO1 in different DNA contexts) though the model proposed in the end contains novel elements regarding the sequential involvement of the exonuclease and endonuclease activities of MRE11 (which is opposite from DSB end resection) and BIR repair of the gap-derived DSB. Nonetheless, both of these novel aspects are speculative and no data in the paper can be found to directly support them.

Respectfully, we in fact found it fascinating that MRE11 and EXO1 cooperate in three different processes (gap expansion, fork degradation and DSB end resection) but in which their activities are differentially employed at distinct steps in those processes. We believe this is a novel concept that may have broad relevance to other fields as well. Moreover, we think this work may stimulate further studies in the field, aimed at understanding the regulation of these enzymes at these various steps. All three processes have great relevance for biology and medicine, thus

understanding their mechanisms and regulation is crucial for developing new drugs and deploying them in the clinic.

The reviewer states that our model on the sequential involvement of the exonuclease and endonuclease activities of MRE11 is speculative and no data in the paper can be found to support it. However, the data presented in Fig. 6 clearly show that both MRE11 exonuclease inhibition, as well as the loss of the exonuclease EXO1, suppress DSB accumulation. This implies that DSB formation cannot occur without gap expansion, and thus that MRE11 endonuclease activity is subsequent to its exonuclease activity during ssDNA gap processing into DSBs.

Regarding the BIR model, as mentioned in our manuscript, this model is indeed just a speculation meant to suggest possible new research directions opened up by our current study. While we are ourselves beginning to address this model in our lab, we believe this is outside the scope of the current manuscript. To remove any confusion, in the revised manuscript we have moved the BIR model figure (Figure 6D in our original manuscript) to the supplementary material (as Supplementary Fig. S9)

2. Piberger et al. (Nature Commun. 2020) reported similar findings that BPDE-induced nascent DNA gaps require MRE11 and EXO1 activities and proposed a similar model that these two nucleases bidirectionally expand PrimPol-generated ssDNA gaps followed by RAD51 binding. Though the DNA replication stressor is different (BPDE vs. HU, BPA, and DEHP), the idea that these two nucleases work together to enlarge the gaps on nascent DNA of the leading strand behind stressed forks is similar.

Indeed, the Piberger et al paper presents a model showing EXO1 and MRE11 engaging a gap at a BPDE adduct site in opposing directions. However, the activity of EXO1 (or of MRE11, for that matter) was not demonstrated in this paper. To our understanding, the only data presented to support their model was the SMART assay to quantify BrdU-labeled ssDNA. However, this assay cannot distinguish between ssDNA at/behind replication forks, or ssDNA formed upon end resection at DSB sites (in fact, this assay was developed as a method to measure end resection at DSB sites -see Altieri et al, *Bio Protoc.* 2020, PMID: 33659366). Thus, even though a schematic model showing an activity of EXO1 in gap expansion was presented, it was not clear from the data presented if the authors were detecting end resection at DSBs or gap expansion. Subsequently, the role of MRE11 in gap expansion was demonstrated using the S1 nuclease DNA fiber assay by the Vindigni lab (Tirman et al, *Mol. Cell* 2021, PMID: 34624216), but the role of EXO1 remain unaddressed until now. To our knowledge, the S1 nuclease fiber assay is the only assay established in the field to be able to measure ssDNA gaps, since the control (no S1 treatment) condition is employed to rule out DSBs. Using this approach, our manuscript is the first to show that EXO1 controls gap expansion. In addition, beyond the genetic experiments with EXO1 inactivation, we show for the first time that EXO1 physically localizes to ssDNA gaps in cells. (For that matter, we show this localization for the first time for MRE11 as well). Thus, our work is the first to demonstrate the role of EXO1 in gap expansion. We believe that, even though aspects of the model may have been speculated before, it is essential to any research field that models are rigorously and unambiguously validated, and sincerely hope that the reviewers agree with this.

Moreover, our model is not just a simple confirmation of the Piberger et al model using other drugs, since it goes beyond, and in a different direction, than this previous model. These authors show that, in response to BPDE treatment, EXO1 activity is required for RAD51 foci

formation, and speculate that a recombination event is occurring downstream, but somehow without DSBs being formed. In contrast, our work in BRCA-deficient cells shows that ssDNA gap expansion is followed by DSB formation, which is toxic to these cells, and defines the molecular events involved in this processing. Thus, our work sheds light on a potentially critical mechanism of DNA damage sensitivity in BRCA-deficient cells.

In the revised manuscript, we present additional results that further extend the relevance of our work. First, we show that MRE11 and EXO1 engage ssDNA gaps in a co-regulated manner (**new Fig. 5a-f, new Supplementary Fig. S7a-c**), highlighting the importance of studying the regulation of this process. Second, we show that loss of EXO1 can partially suppress the HU, cisplatin and BPA sensitivity of BRCA1- or BRCA2-depleted cells (**new Fig. 1h, Fig. 1i, Fig. 6g**), suggesting that ssDNA gap expansion may be partially associated with cellular sensitivity to DNA damaging drugs, at least under the circumstances investigated here.

3. It is unclear how the authors, based on the assays used in the paper, can conclude that the ssDNA gaps observed by the BrdU comet assay and S1 DNA fiber analysis are not on the reversed nascent DNA. It seems that the authors assumed that 0.4mM HU does not induce fork reversal based on the lack of detectable fork degradation phenotype (CldU/IdU ratio) in BRCA-deficient cells. However, Zellweger et al. (JCB, 2015) showed that 1hr treatment of U2OS and RPE-1 cells with 0.5mM HU triggered high frequency (>20%) of fork reversal.

This is indeed an important point, and we thank the reviewer for bringing it up. In the revised manuscript, we addressed this issue at length and include multiple experiments to show that the gaps are not formed on reversed forks. First, we show that depletion of DNA translocases ZRANB3 and SMARCAL1, responsible for fork reversal, does not affect HU-induced ssDNA gap accumulation as analyzed by the S1 nuclease DNA fiber combing assay (**new Fig. 1g**), arguing that MRE11 and EXO1-mediated ssDNA gap formation does not occur on reversed forks. Similarly, depletion of ZRANB3 and SMARCAL1 does not affect ssDNA gap accumulation upon BPA treatment (**new Fig. 4f**), further confirming that these gaps do not form on reversed forks. Finally, and in line with the S1 nuclease experiments, SIRF assays showed that depletion of ZRANB3 or SMARCAL1 does not affect the recruitment of EXO1 or of MRE11 to nascent DNA upon conditions used to induce ssDNA gaps including HU and BPA (**new Fig. 2g, Fig. 2h, Fig. 4j, Fig. 4n, Fig. 5f**). Moreover, in these experiments we also employ siRNA-mediated depletion of EXO1 or MRE11 to confirm the specificity of the SIRF signals observed. Altogether, these findings argue that the gaps processed by EXO1 and MRE11 are not formed on reversed forks.

4. The sample sizes for all the DNA fiber experiments are too small. Even though the combing assay was used, ~50 fibers per condition are not rigorous enough, especially when assessing subtle changes or ruling out effects.

We respectfully disagree with the reviewer in that we would not characterize the differences presented in our manuscript as “subtle”. We believe that the sample size is sufficient since the differences presented are statistically significant as analyzed with the appropriate statistical tests, as established in the literature. The results were reproduced in multiple independent experiments, across multiple cell lines and employing knockdown, knockout and/or chemical inhibition of the factors identified.

Minor points:

1. *Same cell lines should be used to determine the loss of function effects of MRE11 and EXO1 in the contexts of BRCA1 and BRCA2 deficiencies. For instance, in Fig. 1, HeLa cells were used only for BRCA2 loss and RPE1 cells were only used for BRCA2 loss.*

In Figure 1 we used different cell lines for BRCA1 (RPE1) and BRCA2 (HeLa) because of the availability of BRCA-knockout cell lines: BRCA1 knockout was only available to us in RPE1 cells and BRCA2 knockout was only available to us in HeLa cells. We preferred to employ the BRCA-knockout cell lines in order to avoid having to perform co-depletion experiments, which are difficult to control and can give less reliable results due to off-target effects. In the revised manuscript, we extended our analyses using siRNA-mediated depletion of BRCA1 and BRCA2 in the same cell line. To avoid having to perform co-depletion experiments, we first generated EXO1-knockout HeLa cells using CRISPR/Cas9. As presented in the revised manuscript (**new Supplementary Fig. S3a,b**), siRNA-mediated knockdown of BRCA1 or BRCA2 resulted in increased ssDNA gap accumulation upon treatment with 0.4mM HU in wildtype HeLa cells, but not in HeLa-EXO1^{KO} cells. Similarly, inhibition of MRE11 endonuclease activity by the specific inhibitor mirin also suppressed gap formation upon depletion of BRCA1 or BRCA2 in HeLa cells.

2. *Many experiments lack untreated controls to confirm the phenotypes are stress-dependent as the authors claimed. For instance, for all the S1 DNA fiber experiments (Fig. 1D-E, 3J-L, 4C-D) untreated control cells should also be included and analyzed to determine whether the CldU/IdU ratio difference depends on the drugs. It is possible that intrinsic difference in gap formation during unperturbed replication exists between control cells and those lacking BRCA1/2 and/or MRE11 and EXO1.*

In the revised manuscript, we now show that loss of BRCA1/2 and/or MRE11 and EXO1 does not affect the CldU/IdU ratios regardless of S1 treatment, in the absence of genotoxic stress (**new Supplementary Fig. S4a,b**)

3. *In Fig.2B-E, it would be more rigorous to test MRE11 and EXO1 recruitment to nascent DNA using both schemes instead of one scheme for one nuclease only. If the assay can indeed detect the recruitment of the two nucleases to opposite ends of the gaps (5' vs. 3'), this will serve to further test it.*

In the revised manuscript, we provide the experiments requested by the reviewer. As expected, the EXO1 SIRF signal was low when assayed using the MRE11 labeling scheme, which labels the 3' end of the gap (**new Supplementary Fig. S6a**). The other way around, we did detect MRE11 SIRF signal when using the EXO1 labeling scheme, but this is not necessarily unexpected since the 3' gap ends may also be labeled using this scheme (**new Supplementary Fig. S6b,c**).

4. *In Fig.2C and 2F, why did the authors perform statistical analysis in WT cells for EXO1 SIRF but not MRE11 SIRF? In fact, 0.4mM HU in Fig.2C causes visible decrease of MRE11/EdU interaction though no stats are performed. Is this real and if so why?*

Although there is some experimental variation, across multiple experiments we detected no reproducible significant difference in MRE11 or EXO1 SIRF signals in wildtype cells upon exposure to 0.4mM HU (see also the **new Fig. 5f** in the revised manuscript).

5. Based on the DNA fiber data in Fig. 1, it seems 0.4mM HU for 1hr is quite permissive to replication (though slower than untreated condition as inferred from the CldU/IdU ratio). Thus, 0.4mM HU treatment for 3hr (used in many subsequent experiments e.g. Fig.2B) is expected to generate more nascent DNA. If so, how can the authors be sure that the initially EdU-labeled DNA is still “nascent” after the 3hr 0.4mM HU treatment?

We employ the term “nascent” DNA to refer to newly replicated DNA which was labeled with a thymidine analog (and does not necessarily need to be located right at ongoing replication forks). This nomenclature is, to our knowledge, established in the field (see for example Fu et al, *Curr Protoc Cell Biol* 2014, PMID: 25447077).

6. In the abstract and end of introduction, the authors wrote that “the nascent strand” at the ssDNA gap is cleaved by the MRE11 endonuclease generating DSBs. Do the authors mean “template” or “parental” strand?

We apologize for this error and we thank the reviewer for pointing it out. Indeed, we mean the template or parental strand, and not the nascent strand. We fixed this error in the revised manuscript.

7. The MRE11 inhibitor concentrations (e.g. 50uM mirin) seem unnecessarily high.

The mirin concentration used (50uM) is the standard concentration used in the field for these types of assays (see for example Mijic et al, *Nature Commun.* 2017, PMID: 29038466; Quinet et al, *Mol. Cell* 2020, PMID: 31676232). Of note, the Piberger et al paper (PMID: 33203852) indicated above by the reviewer employed an even higher concentration of mirin, namely 100uM.

8. No background information was given as to the chosen concentrations of BPA and DEHP.

We apologize for not including this information in our original manuscript. The studies performed in our manuscript employed 200µM BPA or 200µM DEHP treatment ranging from 30mins to 3h. These treatment conditions were used based on previous literature describing cell culture studies with these compounds (eg 440µM BPA for 2–8 days in PMID: 27208089; 1µM BPA for 3 days in PMID: 20868731; 45µM BPA for 4h in PMID: 27923681; 100µM BPA for 24h in PMID: 33718875; 70µM DEHP for 48h in PMID: 25242624). In the revised manuscript, we now indicate this previous literature (on page 11), and we also comment on the issue of the relevant physiological exposure of organisms to BPA or DEHP (on pages 21-22).

Reviewer #3

This study by Hale et al. investigates the mechanisms through which ssDNA gaps left behind DNA replication forks (RF) are extended by the MRE11 and EXO1 nucleases. The authors also present data regarding the potential of bisphenol A (BPA) and diethylhexyl phthalate (DEHP),

which are used in plastics manufacturing, to generate ssDNA gaps.

We thank the reviewer for their thoughtful comments. Please find below our answers to these comments:

Data indicating that BRCA1/2 prevent the extension of ssDNA gaps by MRE11 and EXO1 is first presented. These results are conceptually consistent with previously published results (e.g., Piberger et al. Nature Communications volume 11, Article number: 5863 (2020)), the difference here being that ssDNA gaps are induced by HU in BRCA1/2-null cells vs bulky adducts. While the data is of good quality, the results were somewhat expected in my opinion.

Indeed, the Piberger et al paper presents a model showing EXO1 and MRE11 engaging a gap at a BPDE adduct site in opposing directions. However, the activity of EXO1 (or of MRE11, for that matter) was not demonstrated in this paper. To our understanding, the only data presented to support their model was the SMART assay to quantify BrdU-labeled ssDNA. However, this assay cannot distinguish between ssDNA at/behind replication forks, or ssDNA formed upon end resection at DSB sites (in fact, this assay was developed as a method to measure end resection at DSB sites -see Altieri et al, *Bio Protoc.* 2020, PMID: 33659366). Thus, even though a schematic model showing an activity of EXO1 in gap expansion was presented, it was not clear from the data presented if the authors were detecting end resection at DSBs or gap expansion. Subsequently, the role of MRE11 in gap expansion was demonstrated using the S1 nuclease DNA fiber assay by the Vindigni lab (Tirman et al, *Mol. Cell* 2021, PMID: 34624216), but the role of EXO1 remain unaddressed until now. To our knowledge, the S1 nuclease fiber assay is the only assay established in the field to be able to measure ssDNA gaps, since the control (no S1 treatment) condition is employed to rule out DSBs. Using this approach, our manuscript is the first to show that EXO1 controls gap expansion. In addition, beyond the genetic experiments with EXO1 inactivation, we show for the first time that EXO1 physically localizes to ssDNA gaps in cells. (For that matter, we show this localization for the first time for MRE11 as well). Thus, our work is the first to demonstrate the role of EXO1 in gap expansion. Moreover, our model is not just a simple confirmation of the Piberger et al model using other drugs, since it goes beyond, and in a different direction, than this previous model. These authors show that, in response to BPDE treatment, EXO1 activity is required for RAD51 foci formation, and speculate that a recombination event is occurring downstream, but somehow without DSBs being formed. In contrast, our work in BRCA-deficient cells shows that ssDNA gap expansion is followed by DSB formation, which is toxic to these cells, and defines the molecular events involved in this processing. Thus, our work sheds light on a potentially critical mechanism of DNA damage sensitivity in BRCA-deficient cells.

I note that several researchers now present only the second label in S1 nuclease assays, rather than the CldU/IdU ratio which can be influenced by RF progression speed prior to exposure to genotoxins. I think it would be beneficial, throughout the manuscript, to present the length of each analog-labeled tracks in addition to the ratio.

In the revised manuscript, for all DNA fiber combing experiments, we provide the length of each individual tracts in the Source Data file. In general, since we compare the effects between -S1 and +S1 conditions, we do not believe that ratios could be affected by genotoxin. In addition, in the revised manuscript we show that, in the absence of genotoxic stress, loss of BRCA1/2 and/or MRE11 and EXO1 does not affect the CldU/IdU ratios regardless of S1 treatment (**new**

Supplementary Fig. S4a,b)

The authors then aim to demonstrate that MRE11 and EXO1 are recruited to ssDNA gaps. In figure 2A, they claim that using 0.4 mM vs 4 mM HU does not cause RF reversal and instead mostly generates ssDNA gaps, based on the absence of “RF protection defect”. In my opinion, these experiments are insufficient to ensure that little/no RF reversal exist in the presence of 0.4 mM HU. First, previous evidence (Zellweger et al. J Cell Biol. 2015 Mar 2; 208(5): 563–579) clearly indicate that 0.5 mM HU causes RF reversal, at least in U2OS cells. Moreover, if I understand the labeling scheme correctly, CldU is not present during the incubation with HU. It is expected that RF will progress in 0.4 mM HU; indeed, data presented in Fig. 1D-E indicate that 0.4mM HU does not completely prevent RF progression. Therefore, RF will progress and may then reverse at some point beyond the CldU labeled track: any RF protection defect in these conditions should be undetectable since it will occur in non-labeled DNA. Even if DNA was still labeled, RF progression would render the detection of eventual RF protection defects difficult since progression and protection defects would counteract each other.

In all our S1 nuclease DNA fiber combing assays, CldU was in fact present during the treatment with 0.4mM HU, as indicated in the labeling schemes presented in each fiber panel. In the absence of S1 treatment, the CldU/IdU ratios were similar between wildtype and BRCA-deficient cells (see for example Fig. 1e, Fig. 1f). Together with the data obtained using the fork degradation labeling setup, also showing no change in the CldU/IdU ratios upon treatment with 0.4mM HU (Supplementary Fig. S5 in the revised manuscript), these findings argue that fork degradation does not occur under these HU treatment conditions.

We thank the reviewer for bringing up the important point of fork reversal. In the revised manuscript, we addressed this issue at length and include multiple experiments to show that the gaps are not formed on reversed forks. First, we show that depletion of DNA translocases ZRANB3 and SMARCAL1, responsible for fork reversal, does not affect HU-induced ssDNA gap accumulation as analyzed by the S1 nuclease DNA fiber combing assay (**new Fig. 1g**), arguing that MRE11 and EXO1-mediated ssDNA gap formation does not occur on reversed forks. Similarly, depletion of ZRANB3 and SMARCAL1 does not affect ssDNA gap accumulation upon BPA treatment (**new Fig. 4f**), further confirming that these gaps do not form on reversed forks. Finally, and in line the S1 nuclease experiments, SIRF assays (requested by the reviewer in their comment below) showed that depletion of ZRANB3 or SMARCAL1 does not affect the recruitment of EXO1 or of MRE11 to nascent DNA upon conditions used to induce ssDNA gaps including HU and BPA (**new Fig. 2g, Fig. 2h, Fig. 4j, Fig. 4n, Fig. 5f**). Moreover, in these experiments we also employ siRNA-mediated depletion of EXO1 or MRE11 to confirm the specificity of the SIRF signals observed. Altogether, these findings argue that the gaps processed by EXO1 and MRE11 are not formed on reversed forks.

Because of the above, in my opinion the SIRF experiments used to claim that MRE11 and EXO1 are recruited to ssDNA gaps, and not to reversed RF, are inconclusive. Indeed, lesser amount of RF reversal followed by nucleolytic activity at the DSB end of the reversed fork might be sufficient to generate, or at least contribute to, the SIRF signals in Figure 2B-G. In my opinion, it is necessary to abolish RF reversal, e.g., using knock-down e.g. of SMARCAL1, and to demonstrate that SIRF signals are still present. Moreover, depletion of PRIMPOL should also reduce the signals in these experiments if indeed they depend on the generation of ssDNA gaps.

As also mentioned above, the SIRF assays requested by the reviewer are now presented in the revised manuscript. Depletion of ZRANB3 or SMARCAL1 does not affect the recruitment of EXO1 or of MRE11 to nascent DNA upon conditions used to induce ssDNA gaps including HU and BPA (**new Fig. 2g, Fig. 2h, Fig. 4j, Fig. 4n, Fig. 5f**), arguing that the gaps processed by EXO1 and MRE11 are not formed on reversed forks. Moreover, EXO1 recruitment to nascent DNA upon treatment with 0.4mM HU is also suppressed by PRIMPOL depletion (**new Fig. 2h**), indicating that EXO1 is recruited to PRIMPOL-mediated gaps. Surprisingly, MRE11 recruitment to nascent DNA upon treatment with 0.4mM HU was not suppressed by PRIMPOL depletion (**new Supplementary Fig. S7c**). The possible causes for this finding are mentioned below in our response to a subsequent comment of the reviewer.

The authors then show that BPA and DEHP causes DNA damage (Figure 3A-I); I note a certain degree of disconnect between these results and the first figures, which seem like a different story.

In the revised manuscript, we show that MRE11 and EXO1 engage ssDNA gaps in a co-regulated manner upon both HU and BPA exposure (**new Fig. 5a-f, new Supplementary Fig. S7a-c**). In addition to providing mechanistic insights into the regulation of MRE11 and EXO1 activities, these studies also serve to better integrate the two parts of our manuscript since the findings are similar with HU and BPA.

While of good quality, these data are not particularly novel as they are in line with several previously published reports (e.g., PMID: 27923681, PMID: 27208089, PMID: 28771232). More novel is the notion that treatment with these agents causes ssDNA gap formation. However, I question the relevance of the doses used; the authors should justify whether these doses are physiologically relevant as they seem quite high. Are these doses close to what happens in organisms affected by BPA/DEHP residues?

We apologize for not including this information in our original manuscript. The studies performed in our manuscript employed 200 μ M BPA or 200 μ M DEHP treatment ranging from 30mins to 3h. These treatment conditions were used based on previous literature describing cell culture studies with these compounds (eg 440 μ M BPA for 2–8 days in PMID: 27208089; 1 μ M BPA for 3 days in PMID: 20868731; 45 μ M BPA for 4h in PMID: 27923681; 100 μ M BPA for 24h in PMID: 33718875; 70 μ M DEHP for 48h in PMID: 25242624). In the revised manuscript, we now indicate this previous literature.

The issue of the relevant physiological exposure of organisms to BPA or DEHP is a complicated one. In one study using *C. elegans*, worms were exposed to 100 μ M and 500 μ M DEHP for several days (PMID: 31917788). In the human population, biomonitoring of BPA has been typically performed from urine samples. Studies of BPA detection in urine found that the median daily excretion of BPA through urine was 34ng/kg of body mass (PMID: 20237498). However, it was subsequently found that BPA accumulates in fat tissues and that biomonitoring studies using urine underestimate the total body burden of BPA (PMID: 22253637; PMID: 12107647). Daily dietary intake of BPA may be as high as 1.5 μ g/kg of body mass (PMID: 27923681; PMID: 23612528). Given that the average weight of an American adult is 80kg, it would take about 377 days to consume 200 μ moles of BPA. Thus, the concentration we employed for our cell culture experiments is higher than the likely physiological exposure of the human population. However, our studies are designed to investigate the acute effects of BPA exposure over control (DMSO) since it is unclear what the exact BPA concentration in various tissues is, and it is impractical to

investigate this under physiological exposure conditions (exposure to lower doses over several years) using these specific assays for measuring ssDNA gap processing. While this is an important caveat of our studies, which we acknowledge in the revised manuscript (pages 21-22) we nevertheless believe that our studies are important to raise awareness to the possible outcomes of BPA exposure, considering how widely spread its usage is, and how human exposure to BPA may only increase in the future.

Importantly, throughout the experiments of Figure 3, the authors mention that BPA and DEHP are particularly deleterious to BRCA-deficient cells; however, no statistical analyses are presented to compare the extent of the effect of the compounds in BRCA1/2-null vs WT cells.

We agree with the reviewer that we did not provide this particular analysis. In the revised manuscript, we provide this analysis for the S1 DNA fiber combing assays (**revised Fig. 3j, Fig. 3k**) since our work focuses on identification of ssDNA gaps upon BPA and DEHP exposure. The novelty of our study is that BPA and DEHP cause gaps even in wildtype cells. While it is not unexpected that BRCA-deficient cells would show increased gap accumulation, since they are deficient in gap repair, we agree with the reviewer that we did not perform specific studies aimed at comparing wildtype to BRCA-deficient cells. We thus carefully revisited how we present this data, toning down the claims that exposure to BPA or DEHP would be particularly deleterious to BRCA mutation carriers.

The authors then present data indicating that MRE11 and EXO1 influence ssDNA gap extension, which is again in line with previously published data using other genotoxins (Piberger et al. Nature Communications volume 11, Article number: 5863 (2020)). In Page 12, the authors state that depletion of MRE11 or of EXO1 suppressed nascent ssDNA accumulation, however, this is not directly shown for EXO1 in my opinion; a BrdU alkaline comet assay (as Figure 4B) or similar experiments would be appropriate.

In the revised manuscript, we now include BrdU alkaline comet experiments obtained with siRNA-mediated EXO1 knockdown, confirming that its depletion also suppresses BPA and DEHP-induced gap accumulation (**new Fig. 4b**). These data further validate the results using the S1 nuclease DNA fiber combing experiments already presented in the original submission (Fig. 4d,e).

The authors go on to use SIRF to state that MRE11 and EXO1 are recruited to ssDNA gaps caused by BPA. However, I think that a similar argument to the one made with regards to 0.4 vs 4 mM HU holds in this case as well. 200 uM BPA clearly permits RF progression (Fig. 4C), and therefore the experiment in Fig. 3G is insufficient in my opinion to state that no RF reversal or degradation occurs in presence of this compound. If RF progress during the incubation with the drugs, then any degradation ("RF protection defect") might be compensated by elongation, rendering the experiment inconclusive. Depletion of PRIMPOL and of RF reversal factors (SMARCAL, ZRANB3, etc...) would provide stronger evidence to link the recruitment of MRE11 and EXO1 specifically at ssDNA gaps and not at reversed forks.

As mentioned above, in our revised manuscript we show that upon BPA treatment: 1) depletion of ZRANB3 or SMARCAL1 does not affect ssDNA gap formation as measured using the S1 nuclease DNA fiber combing assay (**new Fig. 4f**), arguing that the gaps do not occur on reversed forks; and 2) depletion of these translocases does not suppress the recruitment of

MRE11 or EXO1 to nascent DNA as detected by SIF experiments (**new Fig. 4j, Fig. 4n**), arguing that the gaps processed by these nucleases do not form on reversed forks.

Finally, the authors study the possibility that extended gaps might be processed into DSBs by MRE11 endonuclease activity. To this end, they use mirin, but this inhibitor prevents ssDNA gaps extension. While PFM01 inhibits the endonuclease activity of MRE11, it would be important to demonstrate that PFM01 treatment does not influence gap extension in their experimental conditions. Complementation experiments with MRE11 mutants specifically defective in exo vs endonuclease activity (e.g., PMID: 32246962) would also be appropriate.

We agree that investigating separation-of-functions MRE11 mutants specifically defective in exonuclease and endonuclease activities would be very informative. However, this is not straightforward and creating such mutants is not feasible during the course of this a revision. To our knowledge, such mutants have not yet been described in the literature for the human protein. The paper indicated by the reviewer describes a mutant in the *Pyrococcus furiosus* Mre11 protein. Identifying the equivalent mutation in human MRE11 is not straightforward because of the divergence in the protein sequence between the two species, and significant work would need to be performed to confirm that the equivalent human mutant variant retains the characteristics of the archaeal variant described in this paper. Moreover, the paper only describes a mutant that maintain endonuclease activity in the absence of exonuclease activity, but a mutant which retains exonuclease activity in the absence of endonuclease activity has, to our knowledge, not been described so far.

We did perform more work with the MRE11 endonuclease inhibitor as requested by the reviewer. We found that PFM01 treatment partially (but not completely, as compared to the effect of the exonuclease inhibitor mirin) suppressed the ssDNA gaps induced by both HU and BPA in the BrdU alkaline assay (**new Supplementary Fig. S7a,b**), while, at the same concentrations used, PFM01 suppresses DSB induction equally to mirin (Fig. 6b). These findings suggest that MRE11 endonuclease may also be able to engage dsDNA directly and create nicks on nascent DNA, thus complicating the analyses. We further elaborate on this below in our response to a subsequent comment of the reviewer.

I'm also not convinced that the evidence presented is sufficient to claim that MRE11 endonuclease activity acts specifically at ssDNA gaps; it is formally possible that it acts elsewhere, e.g., at stalled RF. More evidence showing that MRE11-induced DSBs occur specifically at ssDNA gaps behind RF are necessary.

We acknowledge the reviewer's comment. Our work shows that :1) MRE11 and EXO1 exonucleolytically extend gaps behind the fork (Fig. 1), and 2) the exonuclease activities of MRE11 and EXO1 are required for DSB formation to occur (Fig. 6). We believe that our proposed model best explains these findings. However, it is indeed formally possible that, in addition to gaps, these exonuclease activities also occur at stalled replication forks (MRE11 on the leading strand and EXO1 on the lagging strand) and it is their engagement at that location, rather than at ssDNA gaps behind the fork, which ultimately causes DSBs. While this seems less likely to us, we nevertheless agree with the reviewer that it cannot be formally ruled out and acknowledge this in the revised manuscript (page 19).

In the revised manuscript, we further explore this aspect. By employing the neutral comet assay, we show that DSB gaps induced in BRCA2-knockout cells by treatment with 0.4mM HU are

suppressed by PRIMPOL depletion (**new Fig. 6a**). This suggest that it is indeed PRIMPOL-derived gaps which are processed into DSBs. In line with this, SIRF experiments showed that EXO1 recruitment to nascent DNA upon treatment of BRCA2-knockout cells with 0.4mM HU is also suppressed by PRIMPOL depletion (**new Fig. 2h**), indicating that EXO1 is recruited to PRIMPOL-mediated gaps. Surprisingly, MRE11 recruitment to nascent DNA upon treatment of BRCA2-knockout cells with 0.4mM HU was not suppressed by PRIMPOL depletion (**new Supplementary Fig. S7c**). We believe this result does not rule out that MRE11 processes ssDNA gaps created by PRIMPOL, but instead it reflects other roles of MRE11 on nascent DNA. This is perhaps in line with the fact that, even though MRE11 and EXO1 co-localize and EXO1 recruitment depends on the presence of MRE11, MRE11 can also be recruited independently of EXO1 (**new Fig. 5a-f**), and also with the fact the MRE11 endonuclease inhibition partially suppresses ssDNA gaps (**new Supplementary Fig. S7c**), overall suggesting that MRE11 has additional functions under these conditions beyond extension of PRIMPOL-generated ssDNA gaps.

We would like to also point out that, as also mentioned above in our response to the reviewer's first comment above, we consider the EXO1 findings as the main aspect of novelty for our studies, since MRE11-mediated exonucleolytic gap expansion was previously shown by the Vindigni lab (Tirman et al, *Mol. Cell* 2021, PMID: 34624216),

It would also be worthwhile to exclude apoptosis-associated DSBs in their experiments, as they cite articles mentioning this possibility in the introduction.

In the revised manuscript, we show that treatment of HeLa-BRCA2^{KO} cells with 0.4mM HU or 200μM BPA for 2 hours does not result in detectable levels of cleaved Caspase-3 (**new Supplementary Fig. S8**). Since neutral comet assays showed that DSBs are formed under these conditions (Fig. 6), this result argues that these DSBs are not apoptosis-associated.

Finally, in their discussion they propose that HR might not be functional at ssDNA gaps-associated DSBs; while this is interesting speculation, they show no data supporting this.

As mentioned in our manuscript, this model is indeed just a speculation, meant to suggest possible new research directions opened up by our current study. While we are ourselves beginning to address this model in our lab, we believe this is outside the scope of the current manuscript. To remove any confusion, in the revised manuscript we have moved the BIR model figure (Figure 6D in our original manuscript) to the supplementary material (as Supplementary Fig. S9)

Minor point:

Page 3: "Arrested forks can also be restarted by initiating de novo DNA synthesis downstream of the lesion, upon repriming by factors such as the primase-polymerase PRIMPOL and, on the leading strand, the Polα-primase complex".

I assume that the authors mean lagging strand here?

We apologize for this error and we thank the reviewer for pointing it out. Indeed, we mean the lagging strand, and not the leading strand. We fixed this error in the revised manuscript.

Page 7: "These findings suggest a general role for EXO1 in ssDNA gap expansion. And for MRE11 presumably since the previous paragraph states essentially the same for this nuclease.

We only mentioned EXO1 here as we wanted to emphasize the impact of EXO1 on gap suppression, which we believe is one of the main novel aspects of our manuscript (while the impact of MRE11 inhibition had been previously addressed by us and others -references 15 and 27 in our manuscript).

REVIEWER COMMENTS

Reviewer #1 (Remarks to the Author):

In their revised manuscript, the authors have altered the manuscript text and have made the manuscript much more nuanced in relation to previous literature. Also the much better connected the two parts of the manuscript (exo1/mre11-mediated processing versus the effects of the environmental agents).

Although the manuscript still describes a somewhat incremental knowledge increase, the new data strengthen the proposed models, and have significantly improved the manuscript. My concerns are also addressed adequately. One point about the representation of data: if all fiber data are reproduced at least once in biologically independent experiments as the authors mention in the rebuttal letter, I would include these data in the figures.

Reviewer #2 (Remarks to the Author):

Major points:

1. In the revised manuscript, the authors provided new data showing that knocking down two well-known DNA translocases SMARCAL1 and ZRANB3 does not prevent the formation of ssDNA gaps in BRCA2-KO cells in response to HU and BPA. They also showed that MRE11 and EXO1 binding on nascent DNA is similarly unaffected by the loss of these two enzymes. Although these new data are consistent with their hypothesis that the ssDNA gaps detected in their experimental system are not on reversed forks, they are unfortunately insufficient to rule this possibility out. First, replication forks can be remodeled/reversed by different enzymes besides SMARCAL1 and ZRANB3. An interesting study by the Cortez lab (Liu et al., Science Advances, 2020) showed that at least two fork remodeling pathways exist to generate fork intermediates which are protected by distinct proteins. While SMARCAL1, ZRANB3, and HLTf work together to reverse forks that are protected by BRCA2, FANCD2, and other proteins, FBH1 reverses forks which require protection from 53BP1, BOD1L, and others. This means that in SMARCAL1 and ZRANB3 knockdown cells, a fraction of the forks can still reverse with the help of FBH1 (and perhaps other unknown enzymes) in response to 0.4mM HU (as shown by Zellweger et al, JCB, 2015). The authors also used the lack of detectable fork degradation in BRCA1/2-depleted cells upon 0.4mM HU treatment as evidence to rule out the existence of reversed forks. Again, this is insufficient as the assay can only detect BRCA1/2-dependent forks which are reversed by SMARCAL1/ZRANB3/HLTf but not those reversed by FBH1 (and possibly other enzymes) and protected by other proteins. In fact, in the new Fig.1g and Fig.4f, the CldU/IdU ratios in cells depleted of ZRANB3 or SMARCAL1 without S1 treatment are clearly higher than the control, indicating that fork reversal is likely occurring to a certain degree during the 1h treatment with 0.4mM HU and 200uM BPA (which slows fork progression and shortens CldU tracts). As such, I do not think the authors have enough experimental evidence in this manuscript to conclude where the ssDNA gaps detected in their system and processed by MRE11 and EXO1 are on the forks. If they are indeed on the reversed forks, the authors have already published the data previously.

2. The finding (Fig.S7) that MRE11 endonuclease inhibitor PFM01 partially suppressed ssDNA gaps induced by HU and BPA is inconsistent with the proposed stepwise model that MRE11 uses its exonuclease activity (with EXO1) to expand ssDNA gaps (earlier) and its endonuclease activity to convert ssDNA gaps to dsDNA breaks (later). Although the authors speculated that MRE11 may play some other unknown roles in ssDNA gap processing on nascent DNA, this finding certainly casts doubt on this particular aspect of their overall model which is a crucial piece to explain how the ssDNA gaps are converted to dsDNA breaks.

3. More evidence than Fig.S8 is needed to rule out apoptosis being a contributor to the dsDNA breaks

detected in their system. The cleaved caspase-3 Western blot does not look very convincing and is not sensitive or quantitative enough to tell if low level of apoptosis is present upon the 2hr drug treatment. The authors should produce higher quality WB data and consider more quantitative apoptosis assays (e.g. by flow cytometry). More importantly, to rule out the contribution of apoptosis, the authors should include apoptosis inhibitors in their dsDNA break detection assays.

Minor points:

1. In Fig.3g, in addition to using siRNAs against BRCA1/2, the authors should also perform the fork degradation assays using the BRCA1 KO RPE1 and BRCA2 KO HeLa cells. Knockdown efficiency may not be as good as the knockout, and it seems important to test the ssDNA gaps and fork degradation under the same experimental conditions as much as possible.
2. Fig.4g and 4k, why did the authors knock down BRCA1 and BRCA2 in BRCA2 KO cells? Wrong labeling?
3. Fig.5f, what is the difference between the first two columns?

Reviewer #3 (Remarks to the Author):

Given the limited time provided for revisions, the authors have addressed my concerns satisfactorily. While overall, I believe the manuscript is now suitable for publication, I do have one lingering concern: the statistical tests used in certain figures or panels seem inappropriate, as corrections for multiple comparisons do not seem to have been considered. Given that many panels feature multiple comparisons, this should be revisited prior to final acceptance, in my view.

Response to referees

Reviewer #1

In their revised manuscript, the authors have altered the manuscript text and have made the manuscript much more nuanced in relation to previous literature. Also the much better connected the two parts of the manuscript (exo1/mre11-mediated processing versus the effects of the environmental agents).

Although the manuscript still describes a somewhat incremental knowledge increase, the new data strengthen the proposed models, and have significantly improved the manuscript. My concerns are also addressed adequately. One point about the representation of data: if all fiber data are reproduced at least once in biologically independent experiments as the authors mention in the rebuttal letter, I would include these data in the figures.

We are glad that the reviewer found that, during the revision process, we have “significantly improved the manuscript” and that the reviewer’s concerns were “addressed adequately”.

In the revised manuscript, we include, in the Source Data file, the results of independent biological replicates for DNA fiber experiments (including Fig. 1e, Fig. 1f, Fig. 4d, Fig. 4e). For the other DNA fiber experiments presented, equivalent samples from independent conceptual replicates were already presented in different panels of the manuscript (eg. Fig. 3j-l and Fig. 1e; Fig. 4d,e and Fig. 3j,k).

Reviewer #2

We thank the reviewer for their comments, and we are grateful to the reviewer for pointing out two errors in the labeling of our figures.

Major points:

1. *In the revised manuscript, the authors provided new data showing that knocking down two well-known DNA translocases SMARCAL1 and ZRANB3 does not prevent the formation of ssDNA gaps in BRCA2-KO cells in response to HU and BPA. They also showed that MRE11 and EXO1 binding on nascent DNA is similarly unaffected by the loss of these two enzymes. Although these new data are consistent with their hypothesis that the ssDNA gaps detected in their experimental system are not on reversed forks, they are unfortunately insufficient to rule this possibility out. First, replication forks can be remodeled/reversed by different enzymes besides SMARCAL1 and ZRANB3. An interesting study by the Cortez lab (Liu et al., Science Advances, 2020) showed that at least two fork remodeling pathways exist to generate fork intermediates which are protected by distinct proteins. While SMARCAL1, ZRANB3, and HLF work together to reverse forks that are protected by BRCA2, FANCD2, and other proteins, FBH1 reverses forks which require protection from 53BP1, BOD1L, and others. This means that in SMARCAL1 and ZRANB3 knockdown cells, a fraction of the forks can still reverse with the help of FBH1 (and perhaps other unknown enzymes) in response to 0.4mM HU (as shown by Zellweger et al, JCB, 2015). The authors also used the lack of detectable fork degradation in BRCA1/2-depleted cells upon 0.4mM HU treatment as evidence to rule out the existence of reversed forks. Again, this is insufficient as the assay can only detect BRCA1/2-dependent forks*

which are reversed by SMARCAL1/ZRANB3/HLTF but not those reversed by FBH1 (and possibly other enzymes) and protected by other proteins. In fact, in the new Fig. 1g and Fig. 4f, the CldU/IdU ratios in cells depleted of ZRANB3 or SMARCAL1 without S1 treatment are clearly higher than the control, indicating that fork reversal is likely occurring to a certain degree during the 1h treatment with 0.4mM HU and 200uM BPA (which slows fork progression and shortens CldU tracts). As such, I do not think the authors have enough experimental evidence in this manuscript to conclude where the ssDNA gaps detected in their system and processed by MRE11 and EXO1 are on the forks. If they are indeed on the reversed forks, the authors have already published the data previously.

The Cortez lab paper indicated by the reviewer (Liu et al., *Sci Adv* 2020, PMID: 33188024) showed that loss of FBH1 does not suppress fork degradation in BRCA2-deficient cells. Since previous studies had shown that fork degradation in BRCA-deficient requires fork reversal (see for example Tagliatela et al, *Mol. Cell* 2017, PMID: 29053959), these findings suggest that FBH1 is not involved in fork reversal in BRCA2-deficient cells. Indeed, in line with this, the Cortez lab paper (Liu et al., *Sci Adv* 2020, PMID: 33188024) shows in fact that BRCA2 is itself required for fork reversal catalyzed by FBH1. Since our S1 nuclease DNA fiber combing experiments for ssDNA gap detection were performed in BRCA2-knockout cells (eg. Fig. 1e,g; Fig. 3j,l; Fig. 4d,f), this rules out FBH1-dependent reversed forks as the structures on which gap expansion occurs.

To confirm this, and experimentally address the reviewer's comment, we depleted FBH1 in BRCA2-knockout cells and measured gap formation upon treatment with 200µM BPA, using the S1 nuclease DNA fiber combing assay. As expected based on the rationale described in the previous paragraph, knockdown of FBH1 did not affect in any way gap induction under these conditions (see below **Figure for Reviewer 2**). These findings confirm that FBH1-mediated fork reversal is not involved in ssDNA gap formation under the experimental conditions investigated here. (We do not believe that these results are relevant to our current work, for the reasons described above, and thus we would prefer to not include them in the manuscript.)

Figure for Reviewer 2. S1 nuclease DNA fiber combing assay showing that FBH1 knockdown does not affect the induction of ssDNA gaps upon treatment of HeLa BRCA2-knockout cells with 200µM BPA. The ratio of CldU to IdU tract lengths is presented, with the median values marked on the graph and listed at the top. Asterisks indicate statistical significance (Mann-Whitney, two-tailed). Schematic representations of the assay conditions are shown at the top.

2. The finding (Fig.S7) that MRE11 endonuclease inhibitor PFM01 partially suppressed ssDNA gaps induced by HU and BPA is inconsistent with the proposed stepwise model that MRE11 uses its exonuclease activity (with EXO1) to expand ssDNA gaps (earlier) and its endonuclease activity to convert ssDNA gaps to dsDNA breaks (later). Although the authors speculated that MRE11 may play some other unknown roles in ssDNA gap processing on nascent DNA, this finding certainly casts doubt on this particular aspect of their overall model which is a crucial piece to explain how the ssDNA gaps are converted to dsDNA breaks.

We understand and appreciate the reviewer's comment. Indeed, the finding that MRE11 endonuclease inhibitor PFM01 partially suppressed ssDNA gaps were unexpected. This experiment was initially requested by Reviewer 3, and we performed it for our initial revision. We reported the results we obtained, and proposed a model to explain them. We believe that, overall, the results presented in our manuscript fit best with the model we proposed in Fig. 7. As explained in our initial rebuttal letter, our speculation that MRE11 has additional functions under these conditions beyond extension of PRIMPOL-generated ssDNA gaps is not solely based on the PFM01 data, but also on our SIRF experiments which showed that: 1) unlike EXO1, MRE11 is also recruited independently of PRIMPOL, and 2) while EXO1 recruitment to gaps is dependent on MRE11, the converse is not true, since MRE11 can also be recruited independently of EXO1.

3. More evidence than Fig.S8 is needed to rule out apoptosis being a contributor to the dsDNA breaks detected in their system. The cleaved caspase-3 Western blot does not look very convincing and is not sensitive or quantitative enough to tell if low level of apoptosis is present upon the 2hr drug treatment. The authors should produce higher quality WB data and consider more quantitative apoptosis assays (e.g. by flow cytometry). More importantly, to rule out the contribution of apoptosis, the authors should include apoptosis inhibitors in their dsDNA break detection assays.

We believe the western blot we presented clearly shows that cleaved Caspase-3 is induced by the control treatment with CPT, but not by treatment with HU or BPA under gap-inducing conditions. Unfortunately, the antibody detects a cross-reactive band that migrates very close to the cleaved Caspase-3 band, and despite our efforts, we were not able to better separate the two bands. In the revised manuscript, we now also measured apoptosis using Annexin V flow cytometry, as requested by the reviewer, and show that, similar to the results of the cleaved Caspase-3 western blot, treatment with 0.4mM HU or 200µM BPA for 2 hours does not induce apoptosis (new **Supplementary Fig. S8b**). Finally, we also performed the experiment with the apoptosis inhibitors requested by the reviewer. We now show that co-treatment with the apoptosis inhibitor Z-VAD-FMK does not suppress DSB induction upon exposure to 0.4mM HU or 200µM BPA for 2 hours (new **Supplementary Fig. S8c**). Altogether, these results rule out the possibility that the DSBs observed under these conditions are a result of apoptosis induction.

Minor points:

1. In Fig.3g, in addition to using siRNAs against BRCA1/2, the authors should also perform the fork degradation assays using the BRCA1 KO RPE1 and BRCA2 KO HeLa cells. Knockdown efficiency may not be as good as the knockout, and it seems important to test the ssDNA gaps and fork degradation under the same experimental conditions as much as possible.

In the revised manuscript, we now present the requested experiment (new **Supplementary Fig. S5b,c**). As expected, treatment with 200 μ M BPA does not cause fork degradation in HeLa-BRCA2^{KO} or RPE1-BRCA1^{KO} cells, similar to the siRNA experiment presented in Fig. 3g.

2. Fig.4g and 4k, why did the authors knock down BRCA1 and BRCA2 in BRCA2 KO cells? Wrong labeling?

This was indeed a labeling mistake. The BRCA1/2 knockdown was done in wildtype DLD1 cells. We thank the reviewer for pointing out this error, for which we apologize. We now fixed the error in the revised manuscript.

3. Fig.5f, what is the difference between the first two columns?

We had forgotten to indicate that the first sample was not treated with HU, while all other samples in the graph were HU-treated. We thank the reviewer for pointing out this additional error, for which we apologize. We now fixed the error in the revised manuscript.

Reviewer #3

Given the limited time provided for revisions, the authors have addressed my concerns satisfactorily. While overall, I believe the manuscript is now suitable for publication, I do have one lingering concern: the statistical tests used in certain figures or panels seem inappropriate, as corrections for multiple comparisons do not seem to have been considered. Given that many panels feature multiple comparisons, this should be revisited prior to final acceptance, in my view.

We are glad that the reviewer found that, during the revision process, we have addressed the reviewer's concerns "satisfactorily", and that the reviewer believes that our revised manuscript is "suitable for publication".

The reviewer comments that our statistical analyses do not include corrections for multiple comparisons. However, we would like to respectfully point out that in our manuscript, we did not perform any multiple comparisons. Even though we are presenting many samples in the graphs, we are only comparing samples pair-wise, as clearly indicated in the graphs. We believe that the pair-wise comparisons indicated in the graphs, as referred to in the text, are the most appropriate for describing the findings and making the conclusions presented in the text, and are, in fact, in line with the literature in the field (eg. Taglialatela et al, *Mol. Cell* 2021, PMID: 34508659; Quinet et al, *Mol. Cell* 2020, PMID: 31676232; Taglialatela et al, *Mol. Cell* 2017, PMID: 29053959). In any case, the Source Data file lists all values plotted in all graphs, thus the reader would be able to perform any statistical analyses of interest.

REVIEWERS' COMMENTS

Reviewer #2 (Remarks to the Author):

In the revised manuscript and rebuttal letter, the authors addressed most of my concerns. The point about whether the ssDNA gaps form on the reversed nascent DNA or not, in my opinion, still remains uncertain despite the new FHB1 knockdown experiment provided in the rebuttal letter. Individually inactivating these reversal enzymes in different pathways is predicted to block the reversal of some, but not all forks. Coupled with the prior study by Zellweger et al. that low dose of HU triggers frequent fork reversal, the authors cannot rule out that some of the ssDNA gaps observed in their experimental system may indeed be on the reversed forks. Since definitive evidence can only be obtained by EM which is non-trivial, the authors should ensure that they do not overstate their findings in the paper and explicitly state the limitation of their approaches and findings. Nonetheless, the manuscript does provide novel insights regarding other regulatory aspects of ssDNA gaps at stressed forks and how they may be rapidly converted to DSB by the endonuclease activity of MRE11 after gap expansion.

Response to referees

Reviewer #2

In the revised manuscript and rebuttal letter, the authors addressed most of my concerns. The point about whether the ssDNA gaps form on the reversed nascent DNA or not, in my opinion, still remains uncertain despite the new FHB1 knockdown experiment provided in the rebuttal letter. Individually inactivating these reversal enzymes in different pathways is predicted to block the reversal of some, but not all forks. Coupled with the prior study by Zellweger et al. that low dose of HU triggers frequent fork reversal, the authors cannot rule out that some of the ssDNA gaps observed in their experimental system may indeed be on the reversed forks. Since definitive evidence can only be obtained by EM which is non-trivial, the authors should ensure that they do not overstate their findings in the paper and explicitly state the limitation of their approaches and findings. Nonetheless, the manuscript does provide novel insights regarding other regulatory aspects of ssDNA gaps at stressed forks and how they may be rapidly converted to DSB by the endonuclease activity of MRE11 after gap expansion.

We thank the reviewer for their comments. We agree with the reviewer that we cannot formally rule out that some of the ssDNA gaps observed in our experimental system occur on reversed forks. In the revised manuscript, we now state this clearly in the Discussion section, on page 19: *“A limitation of our study is that we cannot formally rule out that some of the ssDNA gaps observed occur on reversed forks, considering the multiple fork reversal activities present in cells”*.